# Ending extreme poverty has a negligible impact on global greenhouse gas emissions

Philip Wollburg[1,2 ✉], Stephane Hallegatte[1 ✉] & Daniel Gerszon Mahler[1 ✉]

Growing consumption is both necessary to end extreme poverty[1] and one of the main drivers of greenhouse gas emissions[2], creating a potential tension between alleviating poverty and limiting global warming. Most poverty reduction has historically occurred because of economic growth[3–6], which means that reducing poverty entails increasing not only the consumption of people living in poverty but also the consumption of people with a higher income. Here we estimate the emissions associated with the economic growth needed to alleviate extreme poverty using the international poverty line of US $2.15 per day (ref. 7). Even with historical energy- and carbon-intensity patterns, the global emissions increase associated with alleviating extreme poverty is modest, at 2.37 gigatonnes of carbon dioxide equivalent per year or 4.9% of 2019 global emissions. Lower inequality, higher energy efficiency and decarbonization of energy can ease this tension further: assuming the best historical performance, the emissions for poverty alleviation in 2050 will be reduced by 90%. More ambitious poverty lines require more economic growth in more countries, which leads to notably higher emissions. The challenge to align the development and climate objectives of the world is not in reconciling extreme poverty alleviation with climate objectives but in providing sustainable middle-income standards of living.

Ending extreme poverty requires increasing the consumption levels of all people living above the international poverty line of US $2.15 per day (ref. 7). However, rising income and consumption levels have historically been the main drivers of increasing carbon dioxide equivalent ($CO_2$e) emissions[2]. This raises the question of whether, and under which conditions, containing climate change and alleviating poverty are compatible goals.

Existing research has approached this question by calculating the carbon footprint associated with the consumption of individuals at different income levels using consumption and expenditure surveys[8–10]. These studies simulate increasing the consumption of people living in extreme poverty in the world and estimate the emissions associated with this consumption increase. Studies using this approach have generally found that eradicating poverty leads to modest increases in global emissions, with estimates ranging from less than 1% to about 3%.

Here we approach this question with a different framing. Poverty reduction occurs by a combination of economic growth and distribution of this growth across households, with 90% of historical poverty alleviation driven by economic growth[3–6]. We analyse historical relationships between consumption, economic growth and energy and carbon intensity of gross domestic product (GDP) to estimate the carbon emissions of various growth scenarios under which poverty would be drastically reduced to meet Sustainable Development Goal 1 of ending extreme poverty. With this framing, alleviating poverty requires not only to increase the consumption of people living under the poverty line but also, under realistic assumptions for the distribution of growth based on historical patterns, to increase the consumption of people

not living under the poverty line (Extended Data Fig. 1). Although our objective is not to forecast future growth, poverty or emissions, this approach enables us to assess the emissions implications of poverty alleviation in a range of stylized scenarios under various assumptions for energy and carbon intensities and distributional consequences of growth.

## Growth needed to end extreme poverty

To estimate how much economic growth is needed to end extreme poverty, we first estimate the historical relationship between growth in per capita GDP and growth in per capita consumption in a random slope regression model, taking into account trends across and within countries. We use data for 168 countries from the 2022 Poverty and Shared Prosperity Report of the World Bank[7], converting income distributions to consumption distributions where needed. We find that, on average, when GDP per capita grows by 1%, consumption per capita grows by 0.7%, with variation between countries (Extended Data Table 1a). Using the international extreme poverty line at $2.15 in 2017 purchasing-power-adjusted US dollars, we focus, in our baseline scenario, on the target of reducing the share of people living in extreme poverty to 3% or less—the global poverty reduction target of the World Bank and the interpretation of ending extreme poverty in Sustainable Development Goal 1 of the United Nations[1]. We then estimate the growth necessary to reach this target in each country, assuming an unchanged distribution of consumption within countries. We repeat this exercise for higher poverty lines, $3.65 and $6.85, poverty lines

[1]The World Bank, Washington, DC, USA. [2]Wageningen University and Research, Wageningen, the Netherlands. ✉e-mail: pwollburg@worldbank.org; shallegatte@worldbank.org; dmahler@worldbank.org

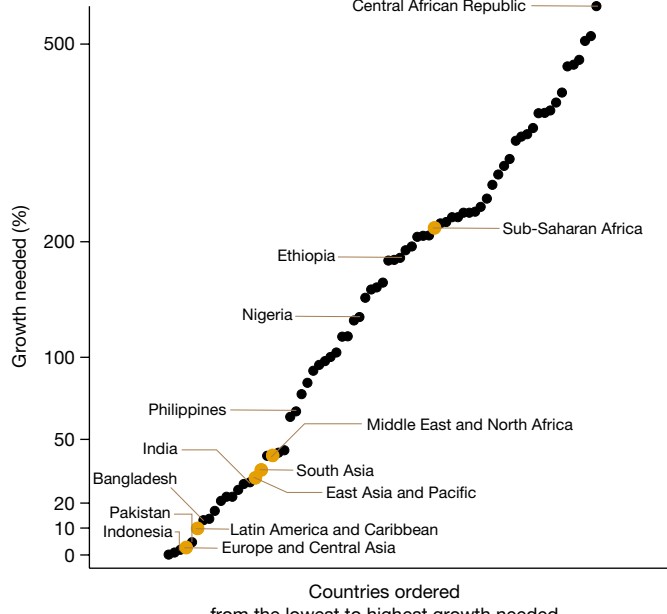

**Fig. 1 | GDP per capita growth needed to reduce extreme poverty to 3%.** All countries with an extreme poverty rate greater than 3% in 2022, each represented with a black dot and ordered according to how much per capita GDP growth it requires to reach a 3% extreme poverty rate. Dots representing the countries of Bangladesh, the Central African Republic, Ethiopia, India, Indonesia, Nigeria, Pakistan and the Philippines are black. Dots in yellow represent regions and show the average growth needed for countries within each region.

typical of lower- and upper-middle-income countries, respectively[11]. More countries would need to grow to alleviate poverty at these higher poverty lines (Extended Data Fig. 2a).

The per capita GDP growth needed to reduce extreme poverty to 3% ranges from 0% to nearly 600% (Fig. 1). Non-poor countries—defined here as countries with extreme poverty rates of less than 3%—require zero growth, as the poverty reduction target is already reached. Targeting the $3.65 and $6.85 poverty lines requires growth between 0% and 1,117% and 0% and 2,251%, respectively (Extended Data Fig. 3).

## Economic growth and emissions

To link GDP growth with greenhouse gas (GHG) emissions, we first relate GDP per capita to energy consumption per capita and then relate energy consumption per capita to GHG emissions per capita. For this, we combine GDP data from the World Development Indicators with data from the Energy Information Administration on primary energy consumption and with data on GHG emissions from Climate Watch, for 2010–2019 (the latest year the data are available).

We again use a random slope regression to model the relationship between GDP and energy consumption. The random slope model exploits both variation between countries and variation within countries and enables countries to convert GDP to energy needs at different rates and improve (or deteriorate) energy efficiency at different rates. The data show a time trend, by which economies have become more efficient across the period we study, at a rate of 1% per year. After accounting for this time trend, a 1% growth in GDP per capita leads, on average, to a 1% increase in energy consumption, although this relationship is different for each country (Extended Data Table 1b and Extended Data Fig. 4a,b). Overall, economies do not become more energy efficient with GDP growth, but they do with time.

We use the same set-up to model the relationship between energy consumption and GHG emissions. We find that for a 1% increase in

energy consumption, GHG emissions grow by 0.7%, with no significant time trend (Extended Data Table 1c and Extended Data Fig. 4c,d). This means that the emissions of the countries grow more slowly than their energy needs, possibly because countries with higher energy consumption are more electrified, which in turn is associated with lower emissions. We also consider non-energy GHG emissions, but find no statistically significant association between GDP growth and non-energy emissions, so we exclude non-energy emissions from the analysis (Extended Data Fig. 4e,f).

## Emissions of poverty alleviation

We now combine these estimates to calculate the carbon emissions needed to reach the extreme poverty reduction target (that is, 3% or less) everywhere. To do so, we compare a counterfactual no-poverty-reduction scenario with a set of illustrative poverty-alleviation scenarios.

The counterfactual no-poverty-reduction scenario keeps consumption distributions unchanged and therefore involves no growth and no poverty reduction. Population grows according to World Bank projections and the estimated historical rates of improvement in energy consumption and carbon intensity hold going forward until 2050.

For the reference poverty-alleviation scenario, all parameters remain the same as in the no-poverty reduction scenario, except for per capita GDP growth, which is calibrated to achieve no more than 3% extreme poverty by 2050. Though the Sustainable Development Goals call for ending poverty by 2030, evidence suggest that this target is out of reach[7]. We select somewhat arbitrarily 2050 as the target year for alleviating poverty and show the sensitivity of this choice in Extended Data Fig. 9b. We extrapolate the current economic growth forecasts into the future in each country until the poverty reduction target is reached. For instance, in India, the 3% target would be met in 2027 based on the current growth trends. Once the target is met in a country, we count only the GHG emissions associated with countries maintaining GDP per capita levels to keep people out of poverty. For countries that are not expected to grow enough to reach the poverty reduction target by 2050, such as Nigeria, we instead model an annualized per capita economic growth rate that meets the 3% target poverty rate in 2050 (Extended Data Fig. 5).

The emissions needed for poverty alleviation are defined as the difference in emissions between the poverty-alleviation scenario and the counterfactual no-poverty-reduction scenario. We count the additional emissions from higher consumption of all people in all countries that have not met the 3% target (including people not living in poverty), not only the additional emissions from people moving out of poverty. In India, for instance, around 6% of the population would need to exit extreme poverty for the target to be reached, but we count the additional emissions from the entire population caused by the economic growth needed to alleviate poverty.

These scenarios are designed to capture the emissions needed to alleviate extreme poverty if historical trends continue. These scenarios do not capture the role of wealthier countries, which have produced the most historical emissions and arguably could do more to weaken the tension between limiting global warming and ending global extreme poverty. We analyse the role of decarbonization in wealthier countries later.

The number of people lifted out of extreme poverty between 2023 and 2050 relative to the no-poverty-reduction scenario amounts to just more than 1 billion by 2050 (Extended Data Fig. 2b). Of the 1 billion, 69% are in sub-Saharan Africa, 19% in South Asia and 5% in the Middle East and North Africa.

Figure 2a shows the emissions associated with meeting poverty alleviation targets at different poverty lines. Annual emissions are estimated to be 2.37 gigatonnes (Gt) of $CO_2$e (or 4.9% of 2019 global emissions) higher in 2050 in the poverty-alleviation scenario than in the no-poverty-reduction scenario. This corresponds approximately

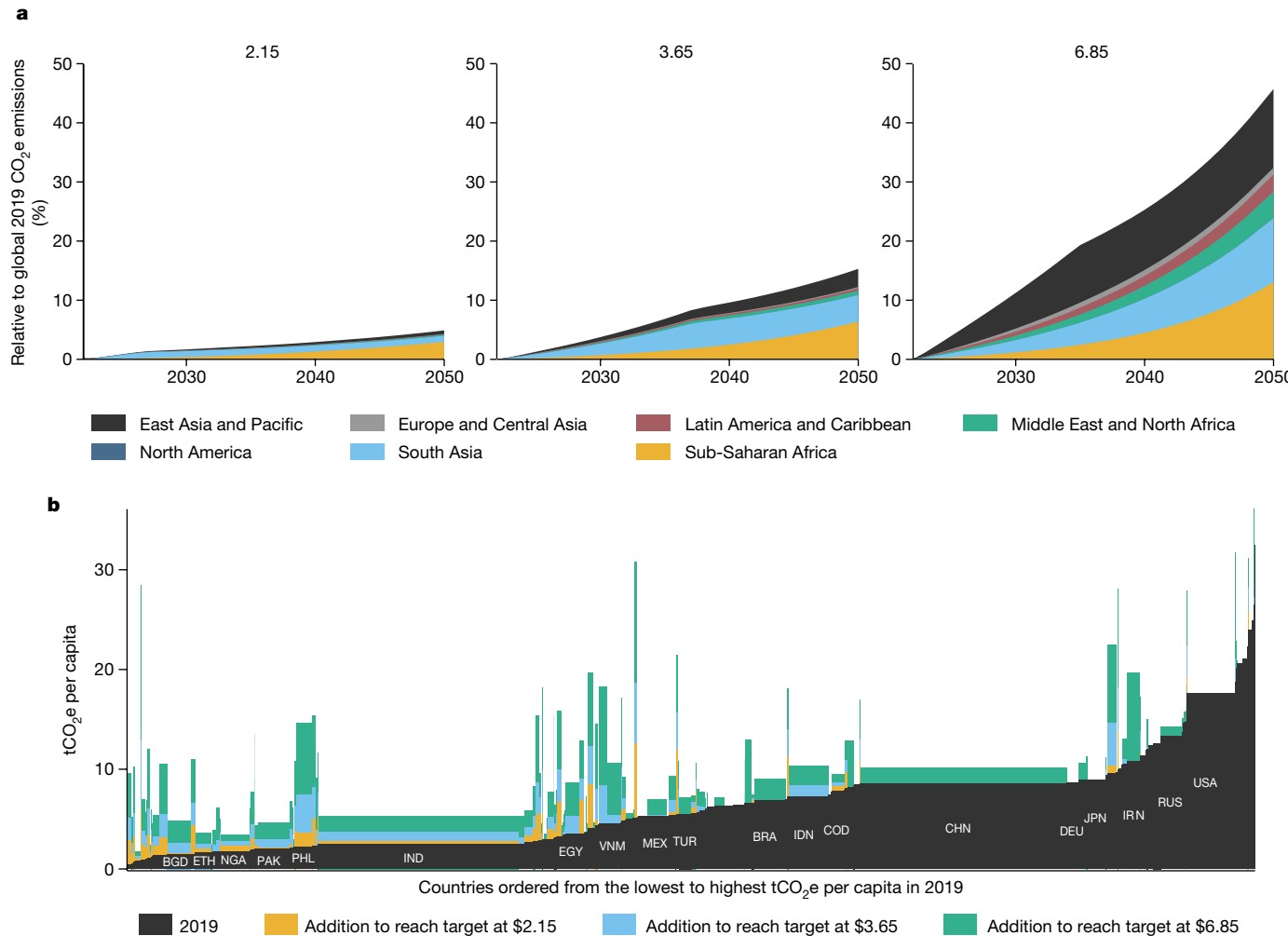

**Fig. 2 | Emissions of poverty alleviation. a,** Annual CO$_2$e increase of poverty reduction at three poverty lines (percentage of 2019 global emissions) by region, **b,** Emissions of poverty alleviation in 2050 by country. The bar width of each country is scaled to their population in 2019. The yellow areas show the CO$_2$e needed to end extreme poverty in 2050, expressed relative to the emissions of the country in 2019. The sum of the blue and yellow areas shows the CO$_2$e needed to reach the target poverty rate of 3% at $3.65, and equivalently for $6.85. tCO$_2$e, tonne carbon dioxide equivalent. BGD, Bangladesh; ETH, Ethiopia; NGA, Nigeria; PAK, Pakistan; PHL, Philippines; IND, India; EGY, Egypt; VNM, Vietnam; MEX, Mexico; TUR, Turkey; BRA, Brazil; IDN, Indonesia; COD, Democratic Republic of the Congo; CHN, China; DEU, Germany; JPN, Japan; IRN, Iran; RUS, Russia; USA, United States of America.

to the increase in emissions the world has been experiencing every 3 years since 2000. The increases in emissions are small in the initial years (0.3% in 2023) and increase over time as more and more people are lifted and kept out of poverty (1.7% in 2030 and 2.9% in 2040). Sixty per cent of the additional emissions in 2050 accrues in sub-Saharan Africa, followed by 21% in South Asia and 12% in East Asia and the Pacific.

Achieving more ambitious poverty reduction targets has more significant consequences on emissions. Using the lower-middle-income poverty line of $3.65 per day triples the increase in annual emissions in 2050 to 7.4 Gt or 15.3% of 2019 global emissions. With the upper-middle-income poverty line of $6.85 per day, the annual emissions in 2050 increase by 22.1 Gt or 45.7%.

The results are relatively modest at the $2.15 line because the emissions of low-income countries are small relative to wealthier countries—even if they reach the income level necessary to meet the 3% target poverty rate with historical energy and carbon intensities (Fig. 2b). By contrast, at the $6.85 line, the added emissions to reach the target poverty rate start to have a notable impact on the global emissions. Twenty-nine per cent of these emissions accrue in each of East Asia and Pacific and sub-Saharan Africa, and 24% accrue in South Asia (Fig. 2a).

## Poverty alleviation and climate change

Even if all new growth would follow historical energy- and carbon-intensity patterns, alleviating extreme poverty does not affect the climate change challenge materially. In the no-poverty-reduction scenario, reaching net-zero GHG emissions in 2050 requires reducing global emissions by 2.0 Gt CO$_2$e per year, factoring in energy and non-energy emissions as well as population growth. In the extreme poverty-alleviation scenario, annual global emissions reduction requirements rise modestly by 4% (from 2.00 to 2.08 Gt CO$_2$e). Non-poor countries (defined as countries with poverty rates of less than 3%) could offset the emissions of poverty alleviation by increasing their historical decarbonization rates by 0.28% per year (Extended Data Fig. 6). Reaching net zero GHG emissions by 2050 is more ambitious than what is needed to keep warming below 1.5 degree with a 50% likelihood and no or limited overshoot[2]. With less stringent objectives (e.g. 10 Gt CO$_2$e in 2050), the result is similar, with the annual global emissions reduction requirement increasing by 5.2% instead of 4%.

Aiming for more ambitious poverty reduction targets creates a more acute trade-off. At the lower-middle-income poverty line of $3.65 per day, the emissions reductions required to achieve net zero by 2050 are 2.14 GtCO$_2$e per year. With the upper-middle-income poverty line

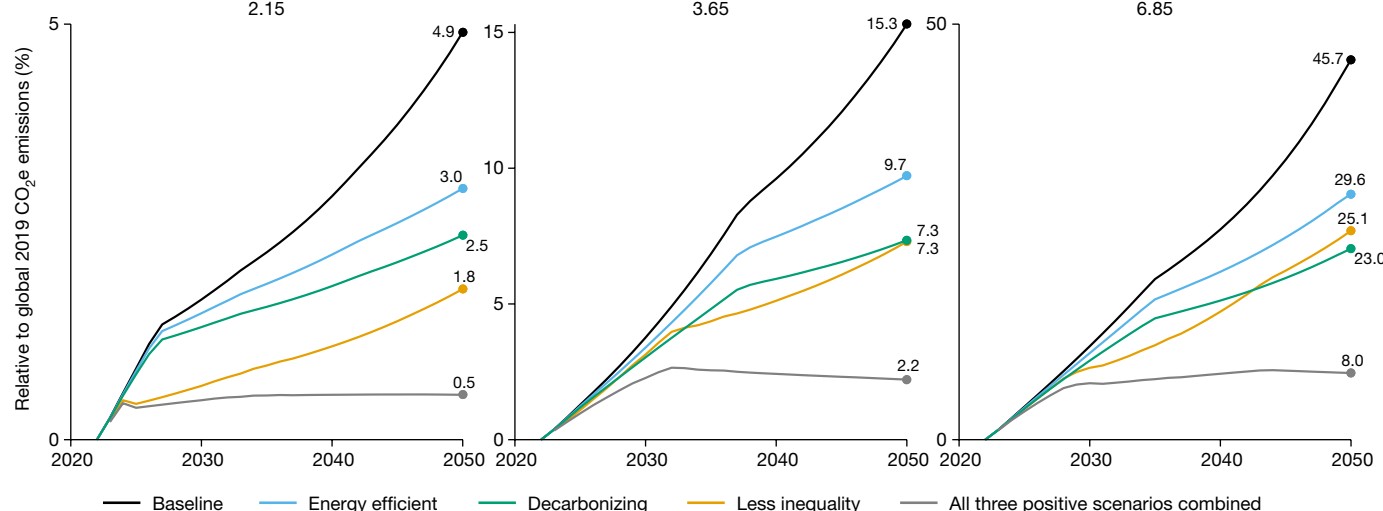

**Fig. 3 | Emissions of poverty alleviation under different scenarios.** Estimated additional annual emissions of poverty alleviation until 2050 for different scenarios. The baseline scenario estimate is shown in black. Comparison scenarios are: low-inequality scenario (yellow); energy-efficient scenario (blue); decarbonization of energy scenario (green); and low inequality, energy efficient and decarbonization scenario combined (grey). Results are shown for three poverty lines: $2.15 per day—the extreme poverty line (left); $3.65 per day—the lower-middle-income poverty line (middle); and $6.85 per day—the upper-middle-income poverty line (right).

of $6.85 per day, the annual global emissions reductions required to achieve net zero rise to 2.42 Gt $CO_2$e between 2023 and 2050.

For comparison, we also calculate the emissions consequences of increasing the GDP per capita of all countries to at least middle-income levels (Extended Data Fig. 7). Raising per capita GDP to the median level of lower-middle-income countries would increase annual emissions in 2050 by 2.0% and by 14.9% to reach the median level of upper-middle-income countries. Because of income inequality, these income levels would not be sufficient to reach the 3% target rate for poverty reduction at the $3.65 and $6.85 lines.

## Inequality, energy and carbon intensity

Changes in inequality matter for the emissions of poverty allevia-tion because they affect the economic growth needed to alleviate poverty[8,12–18]. Although for simplicity our main scenario assumes distributional-neutral growth, we model here a scenario in which coun-tries experience a decline in the Gini coefficient (the most common measure of inequality) at the rate of the top 10% historical Gini declines from 2022 to 2050—a reduction of around 17%. In this scenario, the $CO_2$e emissions increase associated with alleviating extreme poverty in 2050 is 876 million tonnes (Mt) (or 1.8% of 2019 emissions levels)—just more than a third of the 4.9% in the baseline scenario with no inequality change (Fig. 3).

Future economic growth will not have the same energy and carbon intensities as historical patterns. Even without additional climate poli-cies, renewable energies have now become cheaper than fossil fuels in most countries, which will make future growth less carbon intensive than historical patterns[19].

To explore these effects, we consider a scenario in which all coun-tries increase energy efficiency and decarbonize energy consump-tion related to new production at the rate of the top 10% historical performers, a speed of progress roughly similar to the Shared Socio-economic Pathway that is compatible with keeping global warming to 2 °C (ref. 20). In this case, the emissions from poverty reduction in 2050 are reduced to 1.46 Gt or 3.0% of the 2019 emissions, compared with 4.9% in the reference poverty-alleviation scenario. The best historical performance for decarbonization halves the emissions of poverty alleviation in 2050 to 1.19 Gt or 2.5% (Fig. 3).

Combining all three scenarios—lower inequality, energy efficiency and decarbonization of the new production only—brings the emissions of poverty alleviation down to 261 Mt $CO_2$e or 0.54%, a reduction of almost 90% relative to the reference scenario. Combining all three policies would also reduce the additional emissions needed at higher poverty lines: at the $3.65 poverty line from 15.3% to 2.2% and at the $6.85 poverty line from 45.7% to 8.0%.

## Implications for global action

Together these results indicate that the climate challenge cannot be used as a justification for ignoring the people living in extreme poverty in the world. If international organizations, development agencies or governments in low-income countries face trade-offs in policies to mitigate emissions or reduce extreme poverty, alleviating extreme poverty can safely be considered the priority.

The challenge to align the development and climate objectives of the world is not in reconciling extreme poverty alleviation with climate objectives but in alleviating poverty at middle-income standards while containing global warming. This will require decarbonizing the world economy. If governments face trade-offs in policies to mitigate emis-sions or reduce poverty at middle-income lines, there is an urgent need to adopt policies that lower energy intensities, carbon intensities and inequalities.

## Discussion

Our analysis faces several limitations. The modelling framework is deliberately simple to enable transparently comparing different styl-ized scenarios based on historical patterns, rather than attempting to predict the future. The emissions implications of ending extreme poverty may deviate from the results presented here for various rea-sons, but the qualitative findings are robust. Extended Data Table 2 and Extended Data Fig. 8 show a range of results that incorporate deviations from historical patterns and embed the uncertainty of the modelling framework.

More subtle choices, such as the target year for ending extreme pov-erty (2050) and the target poverty rate (3%), also matter for the results (Extended Data Fig. 9a,b). Previous studies, which focused on reducing poverty to 0% without affecting the consumption of wealthier people,

found this to increase global emissions by 2.8% (ref. 10), 1.6–2.1% (ref. 9) and 1.9% (ref. 8). Using a 0% poverty reduction target is not meaningful in our framework because it makes the growth needed to end extreme poverty dependent on the lowest-income households in the country. Because of transitory poverty (for example, because of health shocks) and because of the challenges in measuring the consumption of people living in extreme poverty, using a 0% target makes our results less reliable and unstable, and these results diverge from previous studies as we move closer to 0%. Our framing is also less relevant for very low poverty rates: to eradicate the last pockets of extreme poverty in a country, social protection schemes and redistribution have a stronger role than economy-wide economic growth[21]. In our framework, these transfers are represented as a reduction in inequality and the effects of this reduction on emissions have been explored earlier.

Our framework also does not capture general equilibrium effects or indirect emissions impacts. Global warming is expected to affect poverty levels and may also increase inequality[22], whereas policies to reduce the carbon intensity of growth can also affect inequalities and poverty. GDP growth may also reduce population growth, although accounting for that does not change the results qualitatively (Extended Data Fig. 9c). Accelerated growth in low-income countries may lead to more growth and consequently higher emissions in high-income countries. General equilibrium effects may be particularly relevant for alleviating poverty at higher poverty lines, because they involve larger changes in global consumption and energy use.

Furthermore, monetary welfare measures fail to capture all dimensions of well-being or deprivation[7,23]. Previous research suggested that pathways to end deprivation and satisfying basic human needs can differ from pathways to alleviating monetary poverty and may be achieved at lower emissions intensity[24].

Finally, although we find little trade-off between alleviating extreme poverty and limiting global warming, this does not mean that there are no trade-offs for specific policies or investments. However, recent work points towards more synergies than trade-offs[25] as new technologies and circumstances create new possibilities for pathways that did not exist in the past[26]. For instance, evidence shows that renewable energy sources, rather than fossil fuels, present the most cost-effective way to meet growing electricity demand in many low- and middle-income countries, suggesting that the carbon content of economic growth will be much lower in the future than historically[25]. Particularly relevant for extreme poverty alleviation is the potential from climate-smart agriculture and more efficient land use, as well as small-scale solar mini-grid in rural areas[27]. Also, there is growing evidence of the potential of energy-efficiency measures to generate energy savings and economic benefits, especially linked to the electrification of heat (for example, with heat pumps) and transportation (from electric bikes to electric buses)[28]. When low-energy and low-carbon options become more competitive than alternatives, trade-offs between climate and development objectives disappear, although higher upfront costs and investment needs can represent a major financial challenge.

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

## Methods

### Economic growth needed to end extreme poverty

**Consumption distributions.** Our income and consumption distributions for 2022 come from the 2022 Poverty and Shared Prosperity Report of the World Bank[10], a bi-annual flagship report by the World Bank used for tracking extreme poverty and reporting on the first target of the first Sustainable Development Goal. These 2022 distributions reflect the latest harmonized income or expenditure surveys conducted that the World Bank has access to, extrapolated to 2022 (ref. 29).

Methods and survey designs vary across countries, which may affect comparability[30–32], despite efforts at harmonizing the data across countries. A particular challenge is that the distributions are a mix of consumption and income aggregates. For that reason, we derive a method to convert income (inc) distributions to consumption (con) distributions[33] using the following equation: $con = inc^{0.93} + 0.68 + 0.26 \times \ln(inc_{median})$. Here incomes and consumptions are expressed in daily purchasing-power-parity-adjusted 2017 US dollars.

For about 50 economies home to less than 2% of the population of the world, we have no previous income or consumption data at all. To make the exercise truly global, for these countries we impute consumption using the median value of their respective World Bank income group and region. For region–income group pairs with less than five countries with data, we take the median from the income group of the country with missing data.

Finally, we winsorize the consumption distributions at 50 cents per person per day. Consumption levels below that would reflect a daily caloric intake that is probably impossible to sustain over periods of time, and therefore probably reflect measurement error. This winsorization affects 0.4% of the observations.

We are primarily interested in the GHGs necessary to end extreme poverty, measured at present as falling short of daily consumption of $2.15 in 2017 purchasing-power-parity-adjusted dollars[11]. This is the international poverty line used for the first target of the Sustainable Development Goals and the poverty line used for the mission goal of the World Bank. It reflects the typical national poverty line of low-income countries. These low-income countries tend to define their national poverty lines as the expenditure necessary to consume about 2,200 calories per day and a small non-food allotment.

The $2.15 line is very frugal and individuals with a daily consumption above this threshold may still live in what would ordinarily be considered poverty. To measure the GHGs needed to end poverty at higher thresholds, we also look at the poverty lines typical of lower-middle-income countries ($3.65) and of upper-middle-income countries ($6.85)[11].

We could use national poverty lines, meaning that each country would have its own threshold. However, because national poverty lines are often explicitly or effectively relative in the sense that they increase as countries develop[34,35], there is no reason to believe that poverty according to national standards will ever be ended. Even the wealthiest countries today have poverty according to their national definitions.

### Consumption growth needed to end extreme poverty. With constant distribution.

Calculating the consumption growth necessary to end extreme poverty in each country is straightforward in the case in which growth accrues to all equally—that is, it is distribution neutral. First, we identify the consumption level of the third percentile. Take the case of Benin in which the third percentile reflects a consumption per day of $1.33. For the country to reach the poverty reduction target of the international poverty line in a manner in which the consumption of all individuals grows at an equal rate, the third percentile needs to just pass the poverty threshold. This means that the consumption value of individuals at the third percentile needs to grow by ($2.15 − $1.33)/$1.33 = 62%. As we assume growth is distribution neutral,

the entire consumption distribution of Benin would need to grow by 62% to reach the poverty reduction target. By the same logic, everyone's consumption needs to grow by 174% for the country to reach the target rate for poverty reduction at the $3.65 poverty line. More generally, to reach the target poverty rate of $P^*$ (which unless otherwise specified is 3% in our analysis) at the poverty line $z$, then consumption per capita (growthconpc) in country $c$ needs to grow by

$$growthconpc_c^* = \frac{z}{F_c^{-1}(P^*)} - 1, \tag{1}$$

where $F_c^{-1}(P^*)$ is the consumption level of percentile at $P^*$.

**With changing distribution.** We use the Gini coefficient as the inequality metric because of its popularity. We implement changes in inequality that correspond to taxing consumption by $x$% and distributing the proceeds equally to everyone. This tax and transfer scheme precisely reduces the Gini coefficient by $x$% (refs. 36,37). This particular change in inequality has been shown to occur frequently in historical data[38]. Concretely, it means that if the Gini reduces by $x$%, then each individual's consumption is given by

$$con_{new} = con_{old}(1 - x) + x \times \mu_{con} \tag{2}$$

where $con_{old}$ is the consumption before the inequality change, $con_{new}$ is the consumption after the inequality change and $\mu_{con}$ is the mean consumption per capita.

Given that inequality reductions will increase the consumption of the bottom more than average, the third percentile will now move closer to (or above) the poverty line, so $F_c^{-1}(P^*)$ will increase, and the growth needed for the third percentile to reach the poverty line will be lower. In Benin, the mean consumption is $5.04, so if the Gini is reduced by 10%, the third percentile obtains a consumption level of $1.70, and the growth needed to reach the target drops from 62% to 26%. All of this is shown in Extended Data Fig. 1.

Our baseline scenario uses the distribution-neutral case. Although consumption inequality is expected to change in the coming decades, historical evidence shows that around 90% of changes in poverty are driven by shifts in mean consumption rather than changes in the distribution of consumption[6]. Furthermore, few variables proved helpful in understanding and predicting changes in inequality[3]. Therefore, even if we wanted to try to account for the remaining 10% of historical changes to poverty, it is not obvious how to do so credibly. However, it is possible that distribution neutrality will not hold in the future, for example, if extreme weather events hit the people living in extreme poverty in each country the hardest, leading (in the absence of policy responses) to higher inequality. An alternative method would be to omit growth altogether and link poverty and emissions directly[39].

**GDP per capita growth necessary to end extreme poverty.** Once we know the consumption growth necessary to end extreme poverty, either in the distribution-neutral case or inequality-reducing case, the next step is to convert these consumption growth rates into growth rates in GDP per capita. Evidence from previous studies has shown a discrepancy between consumption growth and GDP growth[40–42]. There are several possible reasons for this, including that part of the GDP growth is saved rather than being allocated to consumption and that GDP growth may be overestimated in some countries[43]. The discrepancy could also be because of unit non-response in surveys or differences in the exact items captured in consumption surveys and national accounts.

To account for this non-one-to-one relationship while acknowledging that the rate at which GDP growth passes through to consumption may differ by country, we fit a random slope model, a variant of what is also known as a multilevel model, a hierarchical linear model or a mixed model[44]. A random slope model is convenient because it exploits both

within- and between-country information. Concretely, we fit a model of the following form:

$$\ln(\text{conpc}_{y,c}) = (\beta_0 + u_{0,c}) + (\beta_1 + u_{1,c})\ln(\text{gdppc}_{y,c}) + \varepsilon_{y,c} \quad (3)$$

Here the $\beta$ parameters are fixed effects constant across countries, whereas the $u$ parameters are country-varying random effects centred around zero. We run this regression on the latest time series of comparable consumption data for each country in the Poverty and Inequality Platform, and match the consumption data with data on GDP per capita from the World Development Indicators supplemented with data from the World Economic Outlook and Maddison database where needed. Extended Data Table 1a shows the regression output.

$\beta_1$ and $u_{1,c}$ are the parameters of interest. $\beta_1$ shows the average rate across countries at which 1% growth in GDP passes through to growth in consumption. $\beta_1$ is estimated to be 0.70 with a standard error of 0.038. $u_{1,c}$ is a country-specific add-on reflecting that the pass-through rate differs by country. The standard deviation of $u_{1,c}$ is estimated to be 0.291 (with a standard error of 0.033). The countries with the largest and smallest $u_{1,c}$ probably reflect historical patterns that are unlikely to replicate. For that reason, we cap $u_{1,c}$ at the 10th and 90th percentile (0.44 and 0.98).

We do not include any time trend in equation (3), unlike in similar regressions that follow to predict energy per capita and GHG per capita (equations (5) and (7)), for three reasons: (1) We have no theoretical prior to suggest a country-specific linear time trend: this country-specific time trend would mean that conditional on a given level of GDP per capita, every year countries continuously increase (or decrease) mean consumption, or equivalently that the savings rate constantly increases (or decreases) without any change to income. (2) If we do include the year in the regression, the fixed effect is highly insignificant. (3) Because the poverty data are not annual in most countries, we have much less power to include country-specific linear time trends.

We can now back out the GDP per capita growth needed to get to the consumption per capita growth required for ending extreme poverty. Benin, for example, is estimated to have a pass-through rate of 0.68. This means that for the 62% consumption per capita growth (calculated earlier) necessary to occur, the GDP per capita needs to grow by 91% (91% × 0.68 = 62%). More generally, the GDP per capita necessary to end extreme poverty is given by

$$\text{gdppc}_c^* = \text{gdppc}_{2022,c}^*[1 + \text{growthconpc}_c^*/(\hat{\beta}_1 + \hat{u}_{1,c})] \quad (4)$$

Here, $\hat{\beta}_1$ and $\hat{u}_{1,c}$ are the estimated parameters from running the regression in equation 3. For the next part of the analysis, it matters when the growth needed to end extreme poverty occurs. In our baseline set-up, we use growth forecasts in real GDP per capita from the October 2022 World Economic Outlook of the International Monetary Fund. These growth forecasts continue only until 2027, beyond which we assume that the growth rate for 2027 continues onwards to 2050. If countries have not reached $\text{gdppc}_c^*$ by 2050, instead of using International Monetary Fund growth forecasts, we assign countries the annualized growth rate needed to exactly reach $\text{gdppc}_c^*$ by 2050. We do so because some countries are not on track to reach the target GDP level any time soon, and modelling many decades ahead would add to the uncertainty of the results. In Extended Data Fig. 9b, we show how the assumption of all countries ending extreme poverty by 2050 matters for our results.

### Economic growth and GHG emissions
Once the GDP per capita growth necessary to end extreme poverty is estimated for each country, we calculate the GHG emissions associated with this growth. We consider GHGs from energy and non-energy separately.

**Energy levels.** For energy emissions, we take the intermediate step of first modelling the energy levels. This has the advantage of enabling us to separately explore the impact of energy intensity of GDP and the impact of carbon intensity of energy. The energy data are drawn from the US Energy Information Administration and cover primary energy consumption.

Extended Data Fig. 4a,b shows energy per capita as a function of GDP per capita across countries in 2019, the latest year with data at the time of writing, and shows the cross-country fit over the past two decades. Countries with a higher GDP per capita use more energy per capita, and the energy needs for a given level of GDP have decreased over the past two decades. To fit a model to these stylized facts, we once again run a random slope model, this time enabling country variation in how GDP per capita is converted to energy needs and in how energy needs change over time by adding a year variable. This enables countries to produce the same GDP with less energy year by year, and for this rate of improvement in energy intensity to vary by country.

We run the model with data from 2010 onwards, as older data may contain patterns that are less relevant for the future. Occasionally, there are clear breaks in the energy data series, which, if ignored, would give unreliable predictions. We identify breaks by calculating the average annual change in energy consumption per capita by country, and flag whenever an annual change is more than four times the average change for a country. Whenever a break is identified, we use only data after the break. Equation (5) shows the regression we run and Extended Data Table 1b the results of the regression.

$$\ln(\text{energypc}_{y,c}) = (\beta_0 + u_{0,c}) + (\beta_1 + u_{1,c})\ln(\text{gdppc}_{y,c}) \\ + (\beta_2 + u_{2,c})\text{year} + \varepsilon_{y,c} \quad (5)$$

On average, a 1% growth in GDP leads to a 1% growth in energy, but this effect varies greatly across countries, with the standard deviation being 0.33%. Every year, countries, on average, get 0.9% more efficient at producing the same level of GDP, but again there is large country variation, with the standard deviation of the random effect being 2.5%.

As was the case for the prediction of GDP levels, we once again trim the country-level distributions of random effects at the 10th and 90th percentiles, which is 0.78% and 1.27% for GDP per capita and −3.2% and 2.1% for the annual change. We do so because the most extreme historical patterns are unlikely to continue in the future. Moreover, we also identify the most extreme outliers in the relationship between energy per capita and GDP per capita (evaluated as the residual from the linear trend line in 2022) and shift those towards the trendline so the residual does not exceed the 10th and 90th percentiles in the distribution of residuals.

Based on these, we can predict the target energy per capita level needed to end extreme poverty in 2050 as

$$\ln(\text{energypc}_{c,2050}^*) = (\hat{\beta}_0 + \hat{u}_{0,c}) + (\hat{\beta}_1 + \hat{u}_{1,c})\ln(\text{gdppc}_{c,2050}^*) \\ + (\hat{\beta}_2 + \hat{u}_{2,c}) \times 2050 \quad (6)$$

**Energy GHG emissions.** Next, we convert these energy predictions to predictions of GHGs from energy. Extended Data Fig. 4c,d shows the cross-country relationship and how it has changed over time. There is clear evidence of larger energy needs leading to more energy GHGs, but little evidence of countries improving their ability to produce energy levels with fewer GHGs over time.

Our approach to model these patterns is identical to the one followed above: we once again run a random slope model, this time predicting energy emissions as a function of time and energy levels while allowing for cross-country heterogeneity. The regression we run is listed in equation (7) and the output is presented in Extended Data Table 1c. We limit the impact of outliers in the same way as for the regression of GDP per capita on energy per capita.

$$\ln(\text{ghgenergypc}_{y,c}) = (\beta_0 + u_{0,c}) + (\beta_1 + u_{1,c})\ln(\text{energypc}_{y,c}) \\ + (\beta_2 + u_{2,c})\text{year} + \varepsilon_{y,c} \tag{7}$$

The regression output confirms the visual pattern from Extended Data Fig. 4c,d. Higher energy per capita leads to higher GHGs from energy, and there is no evidence of decreased carbon intensity of energy over time. The latter might seem counterintuitive given that the share of renewable energy of total energy has increased over time. Yet rather than being picked up by the time coefficient, this effect is being picked up by the coefficient on energy per capita, which is less than 1% on average. When energy per capita increases by 1%, GHGs per capita from energy on average increase by only 0.69%.

**Non-energy GHGs.** With the same methodology, we do not find any statistically significant association between GDP growth and non-energy emissions, and exclude it from the calculation of the GHGs associated with ending extreme poverty. Taken at face value, this means that based on historical data, we should not expect non-energy GHGs per capita to increase as the economy of a country grows. Although there may be exceptions to this pattern, such as countries for which non-energy GHGs have increased systematically as a country developed because of deforestation, Extended Data Fig. 4e,f suggests that for any country in which this happened, there is another country in which the reverse happened.

### GHGs to end extreme poverty
With the modelling above, we can estimate the annual GHGs as at present poor countries (defined as countries with poverty rates greater than 3%) approach the GDP per capita necessary to end extreme poverty, which we refer to as the poverty-alleviation scenario. For 2050, this equals

$$\ln(\text{ghgenergypc}^*_{c,2050}) = (\hat{\beta}_0 + \hat{u}_{0,c}) + (\hat{\beta}_1 + \hat{u}_{1,c})\ln(\text{energypc}^*_{c,2050}) \\ + (\hat{\beta}_2 + \hat{u}_{2,c}) \times 2050 \tag{8}$$

Some of these GHGs would also be emitted even if poor countries made no progress in eliminating poverty. To quantify the additional GHGs necessary to end extreme poverty, we need a counterfactual scenario. To that end, we calculate annual GHG emissions for each poor country if they do not grow their GDP per capita beyond their current level: $\text{gdppc}^0_c = \text{gdppc}_{2022,c}$. We can fit this in equation (6) to obtain $\ln(\text{energypc}^0_c)$, which we then fit into equation (8) to obtain $\ln(\text{ghgenergypc}^0_c)$ —the GHGs we would expect from the country if it does not grow until 2050 but otherwise follow the same patterns as in our poverty-alleviation scenario. We call this the no-poverty-reduction scenario.

Each year, the difference between the poverty-alleviation and no-poverty-reduction scenarios shows the additional GHGs needed for the country to be on the path to alleviate extreme poverty. In 2050, it shows the additional GHGs needed for the country to alleviate extreme poverty and will be calculated as

$$\text{ghgneeded}_{c,2050} = (\text{ghgenergypc}^*_{c,2050} - \text{ghgenergypc}^0_{c,2050}) \\ \times \text{pop}_{c,2050} \tag{9}$$

where $\text{pop}_{c,2050}$ is the population of country $c$ in 2050 according to World Bank population forecasts. In an alternative scenario, we model population growth as endogenous to our model, which affects our results only marginally (Extended Data Fig. 9c).

For the countries that are projected to end extreme poverty before 2050, the poverty-alleviation scenario grows economies just until the point at which they have reached the poverty reduction target, and after that keeps it constant. Once the poverty reduction target is reached,

this means that we estimate the GHGs necessary to maintain a GDP per capita to maintain the target poverty rate.

To calculate the total global GHGs needed to reach the poverty reduction target, we simply sum over all countries that had not reached the poverty reduction target in 2022 ($C_{\text{poor}}$):

$$\text{ghgneeded}_{\text{world},2050} = \sum_{c \in C_{\text{poor}}} \text{ghgneeded}_{c,2050} \tag{10}$$

### Alternative scenarios
We use as best historical performances in the energy efficiency of GDP and the carbon intensity of energy the 10th percentile of the distributions of random coefficients for all countries. This scenario assumes annual improvements in energy efficiency of 3.2% and in carbon efficiency of 2.1% per year. This is similar to SSP1–26, which globally assumes annual improvement of 3.4% and 2.4% (ref. 20).

To model changes in inequality, we first need to derive the distribution of inequality changes observed historically. The distribution of inequality changes depends on the time period analysed—year-to-year changes tend to be smaller than changes observed over a decade. We look at all inequality changes observed in the Poverty and Inequality Platform of the World Bank and plot them as a function of the time between the estimates. For each time period between estimates, the 10th percentiles of the distribution of changes in the Gini are considered as our inequality reduction scenario.

Using all available historical data, over a 16-year period, the 10th-percentile inequality change is equal to a reduction of the initial Gini of 17%. There are 25 or fewer comparable Gini estimates 17 years apart (or more), which we use as a minimum for calculating the distribution of inequality changes. We are interested in inequality changes occurring over 28 years (from 2022 to 2050) and assume that inequality does not change further after 16 years.

Note that for each of these scenarios, we change not only the poverty alleviation scenario but also the counterfactual no-poverty alleviation, in line with our approach of strictly isolating emissions from all per capita GDP growth needed to end extreme poverty. In the Supplementary Information, we explore further scenarios tweaking population growth rates, GDP-to-consumption pass-through rates and worst historical performers.

### Data availability
The datasets generated during and/or analysed during the current study are available at Zenodo (https://doi.org/10.5281/zenodo.8275494). Source data are provided with this paper.

### Code availability
The code used to generate the analysis is available at Zenodo (https://doi.org/10.5281/zenodo.8275494).

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

**Acknowledgements** We are thankful for the comments received from S. Freije-Rodriguez, G. Carletto and the participants at a presentation to the members of the Poverty and Equity Global Practice of the World Bank and to the members of the Global Poverty and Inequality Data team of the World Bank. D.G.M. acknowledges financial support from the UK government through the Data and Evidence for Tackling Extreme Poverty (DEEP) Research Programme and from the World Bank Group SDG Fund. The findings, interpretations and conclusions expressed in this paper are entirely ours. These findings do not necessarily represent the views of the World Bank and its affiliated organizations, or those of the Executive Directors of the World Bank or the governments they represent.

**Author contributions** The authors randomized the author ordering (P.W., S.H. and D.G.M.). All authors contributed to the paper. P.W. contributed to the conceptualization, methodology, formal analysis and writing of the original draft, review, editing and visualization. S.H. contributed to the conceptualization, methodology, writing, review, editing, supervision and guidance. D.G.M. contributed to the conceptualization, methodology, formal analysis, data curation, software, writing of the original draft, review, editing and visualization.

**Competing interests** The authors declare no competing interests.

**Additional information**
**Correspondence and requests for materials** should be addressed to Philip Wollburg, Stephane Hallegatte or Daniel Gerszon Mahler.

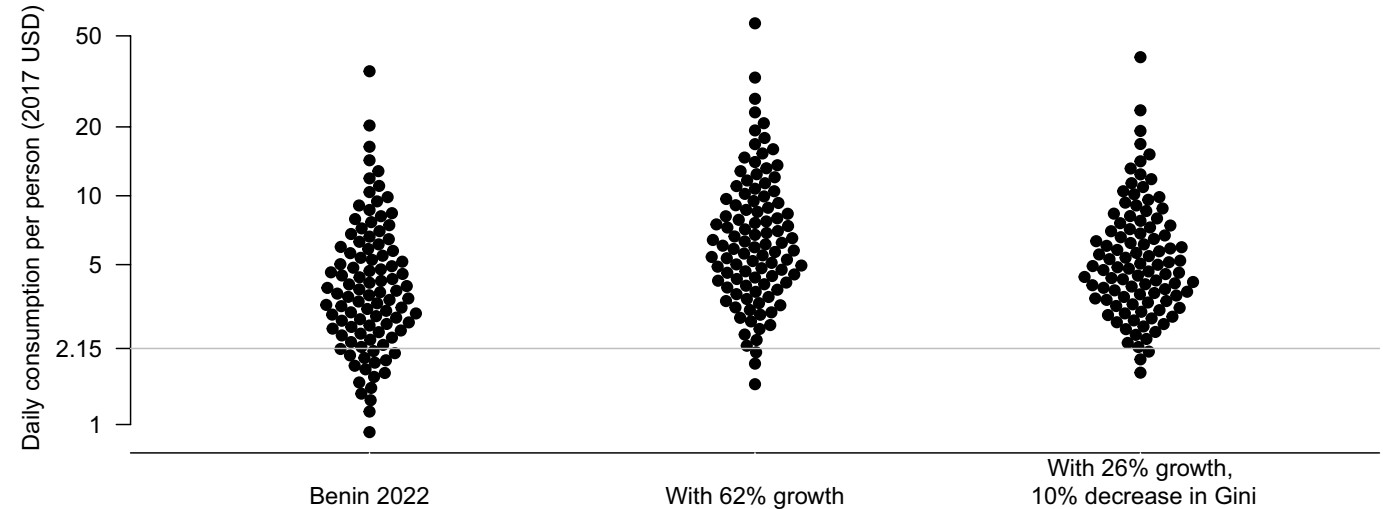

**Extended Data Fig. 1 | Consumption growth necessary to reach the poverty reduction target, Benin.** The left part shows the distribution of consumption in Benin boiled down to 100 percentiles. The middle and right part show examples of growth and redistribution that can make Benin reach the 3% target poverty rate.

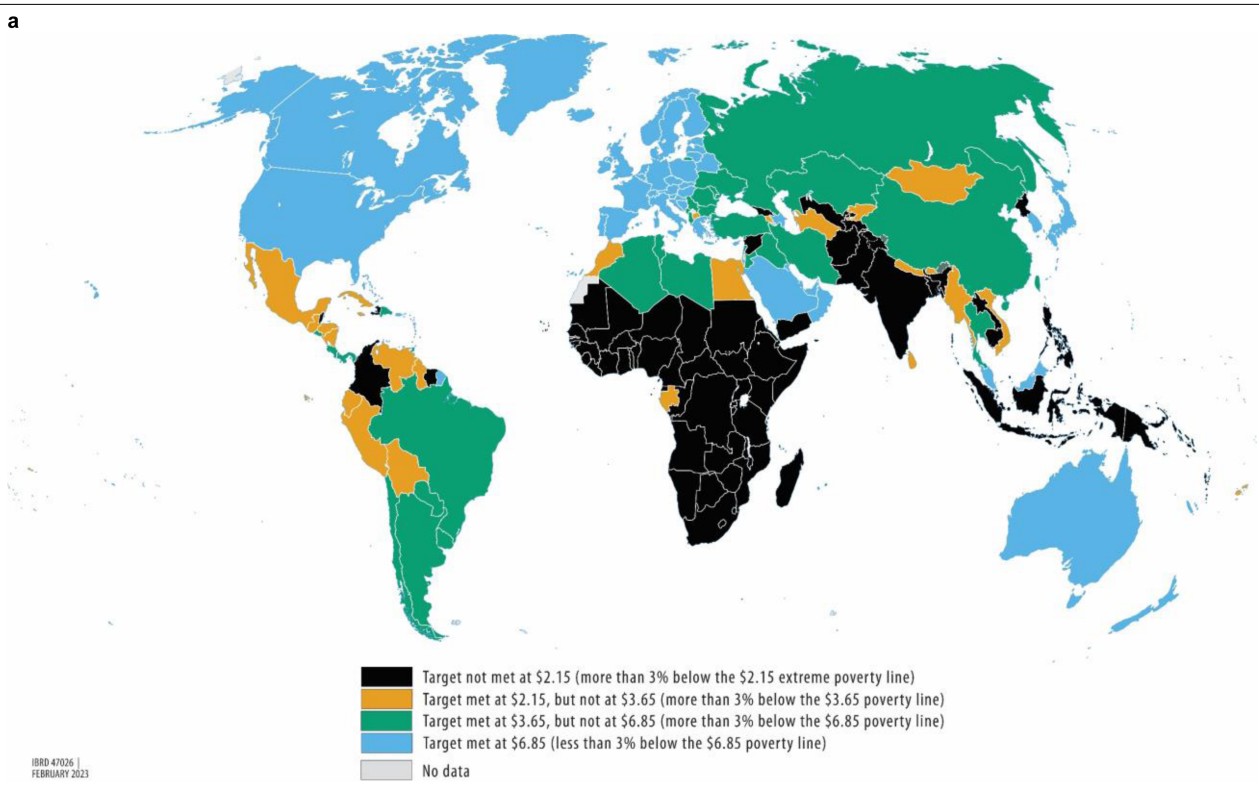

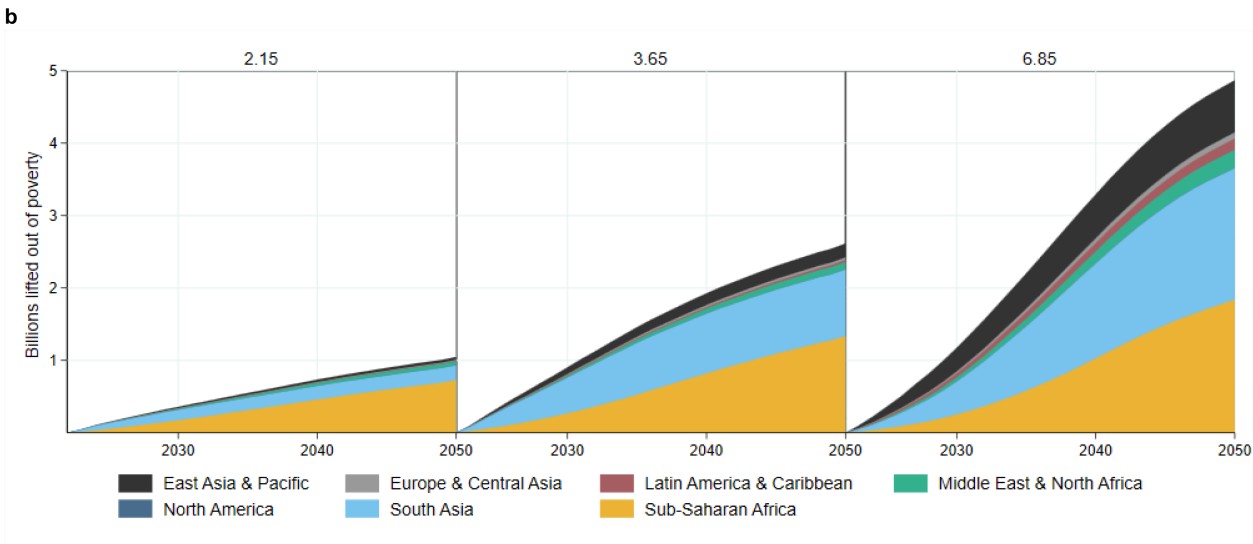

**Extended Data Fig. 2 | Poverty levels and people to lift out of poverty to reach poverty reduction target. a**. Categorization of countries by whether they have met the 3% target poverty rate. Uses consumption distributions for 2022. **b**, People lifted out of poverty for poverty to reach 3% by 2050.

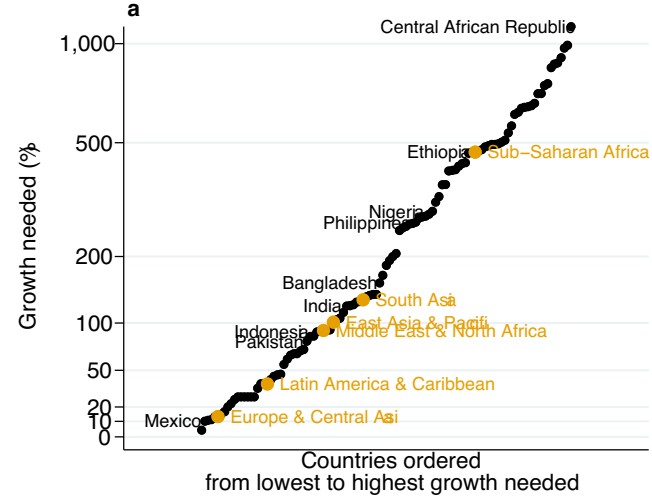
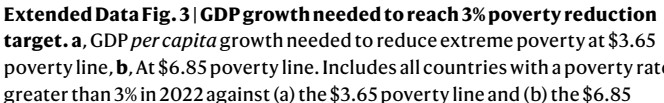
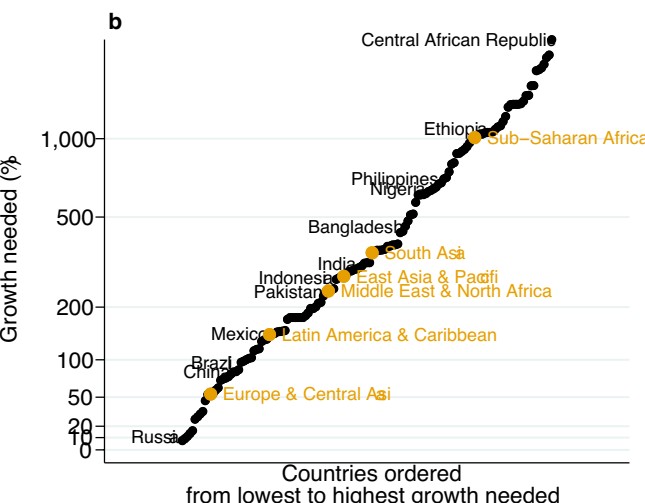

**Extended Data Fig. 3 | GDP growth needed to reach 3% poverty reduction target. a**, GDP *per capita* growth needed to reduce extreme poverty at $3.65 poverty line, **b**, At $6.85 poverty line. Includes all countries with a poverty rate greater than 3% in 2022 against (a) the $3.65 poverty line and (b) the $6.85 poverty line. Each country represented by a black dot. Yellow labelled dots represent regions and show the average *per capita* GDP growth needed for all countries within a given region.

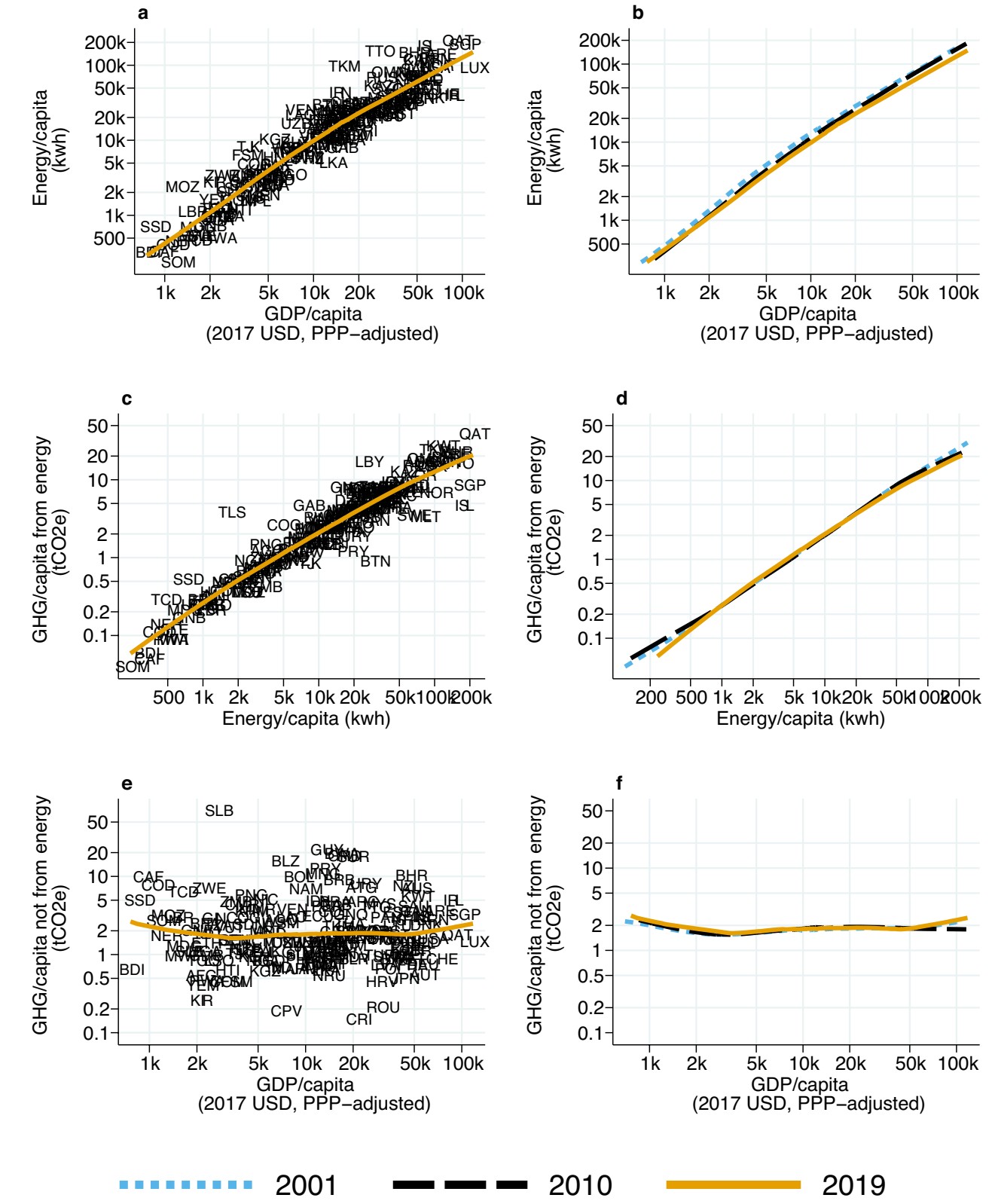

**Extended Data Fig. 4 | Relationship between GDP *per capita*, energy *per capita*, and greenhouse gases *per capita*. a**, Cross-country relationship between GDP *per capita* and energy *per capita* in 2019, **b**. Cross-country relationship between GDP *per capita* and energy *per capita* over time. **c**, Cross-country relationship between energy *per capita* and greenhouse gases from energy *per capita* in 2019, **d**, Cross-country relationship between energy *per capita* and greenhouse gases from energy *per capita* over time. **e**, Cross-country relationship between GDP *per capita* and non-energy greenhouse gases *per capita* in 2019, **f**, Cross-country relationship between GDP *per capita* and non-energy greenhouse gases *per capita* over time. k = 1,000. Source: World Development Indicators, World Economic Outlook, the Maddison Project Database, the U.S Energy Information Administration, and ClimateWatchData (CAIT).

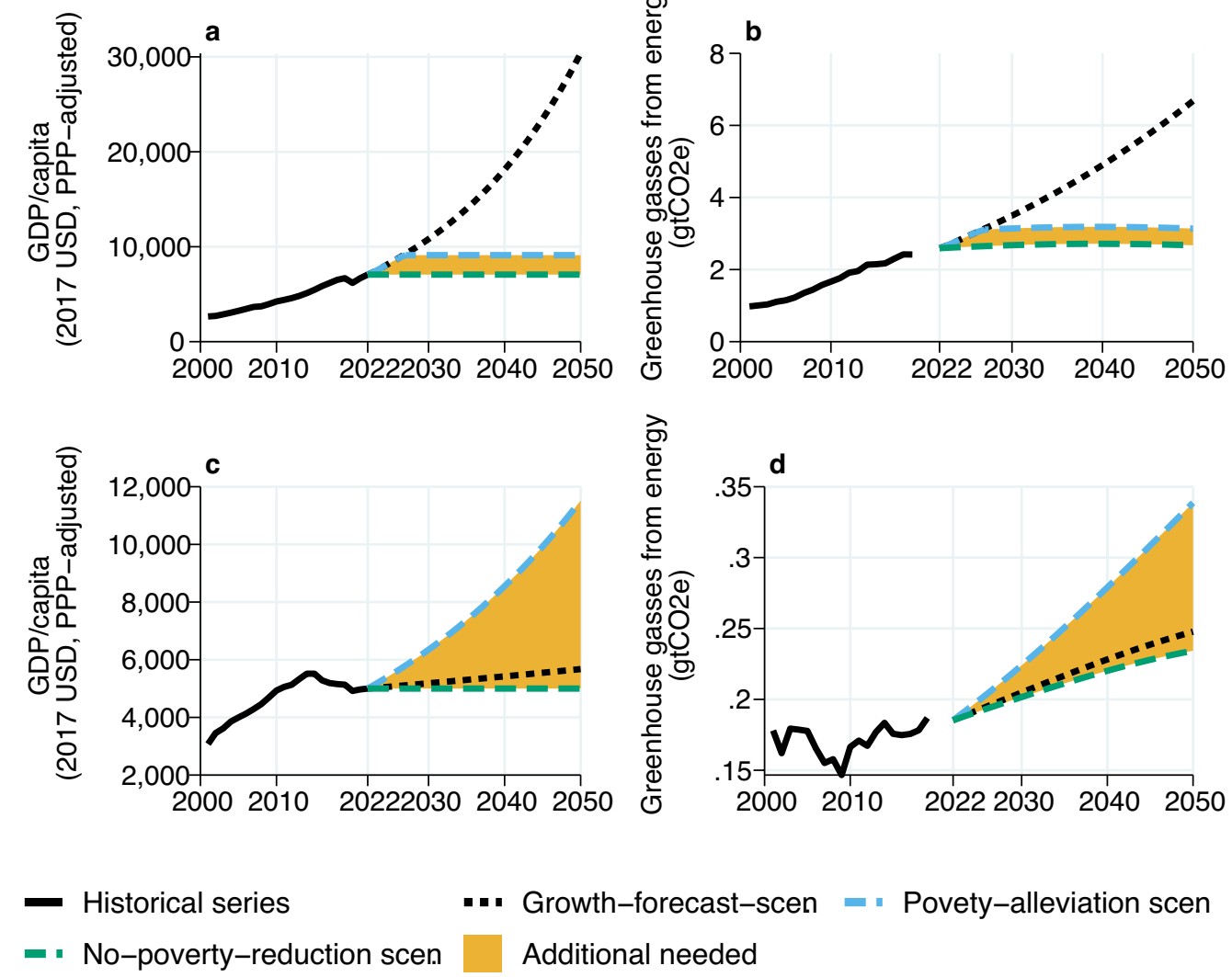

**Extended Data Fig. 5 | Illustrations of GDP *per capita* and greenhouse gases from energy needed to end extreme poverty. a**, GDP *per capita* in India, **b**, CO2e from energy in India, **c**. GDP *per capita* in Nigeria, **d**, CO2e from energy in Nigeria: All panels show the poverty-alleviation scenario and no-poverty-reduction scenario, in which poor countries do not grow beyond 2022. The yellow area is the additional GDP/capita or greenhouse gases needed to end extreme poverty. The figures also include a growth-forecast-scenario, which shows the GDP and greenhouse gases towards 2050 if the countries grow according to IMF growth expectations, which may be more or less than the growth needed to end extreme poverty.

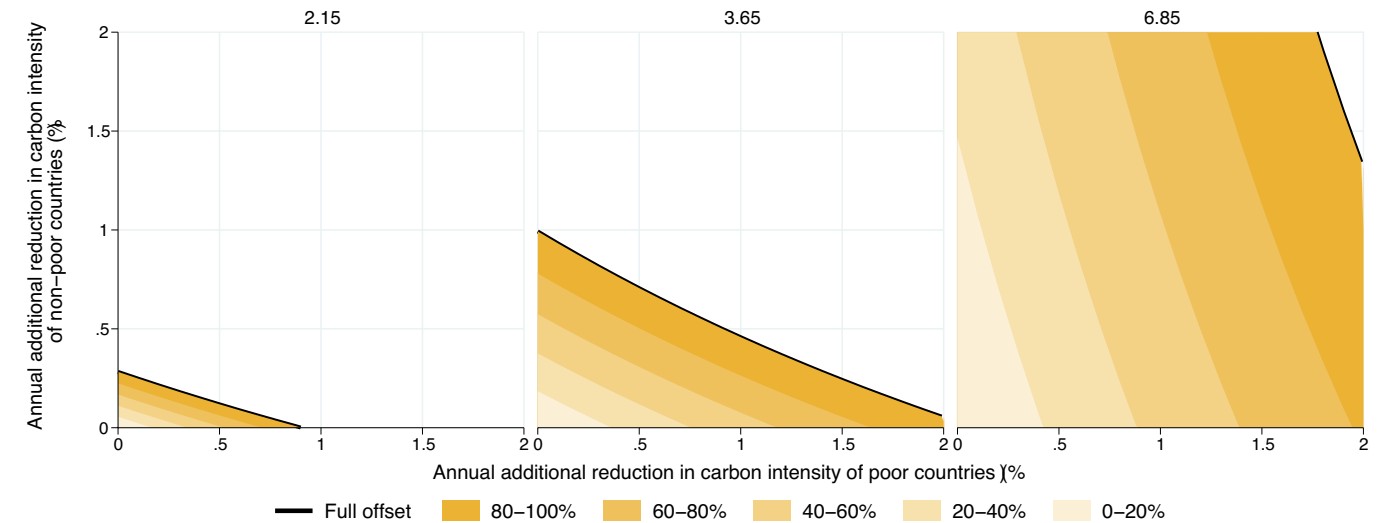

**Extended Data Fig. 6 | ISO-GHG curve of offsetting the emissions of poverty alleviation in non-poor and poor countries.** Black lines show all combinations of reductions in carbon intensity in poor and non-poor countries that would offset entirely the emissions from poverty alleviation. The intersection with the vertical axis is the reduction needed if coming from non-poor countries alone, while the intersection with the horizontal axis is the reduction needed if coming from poor countries alone. Countries are defined as poor if they have more than 3% poverty at a given poverty line.

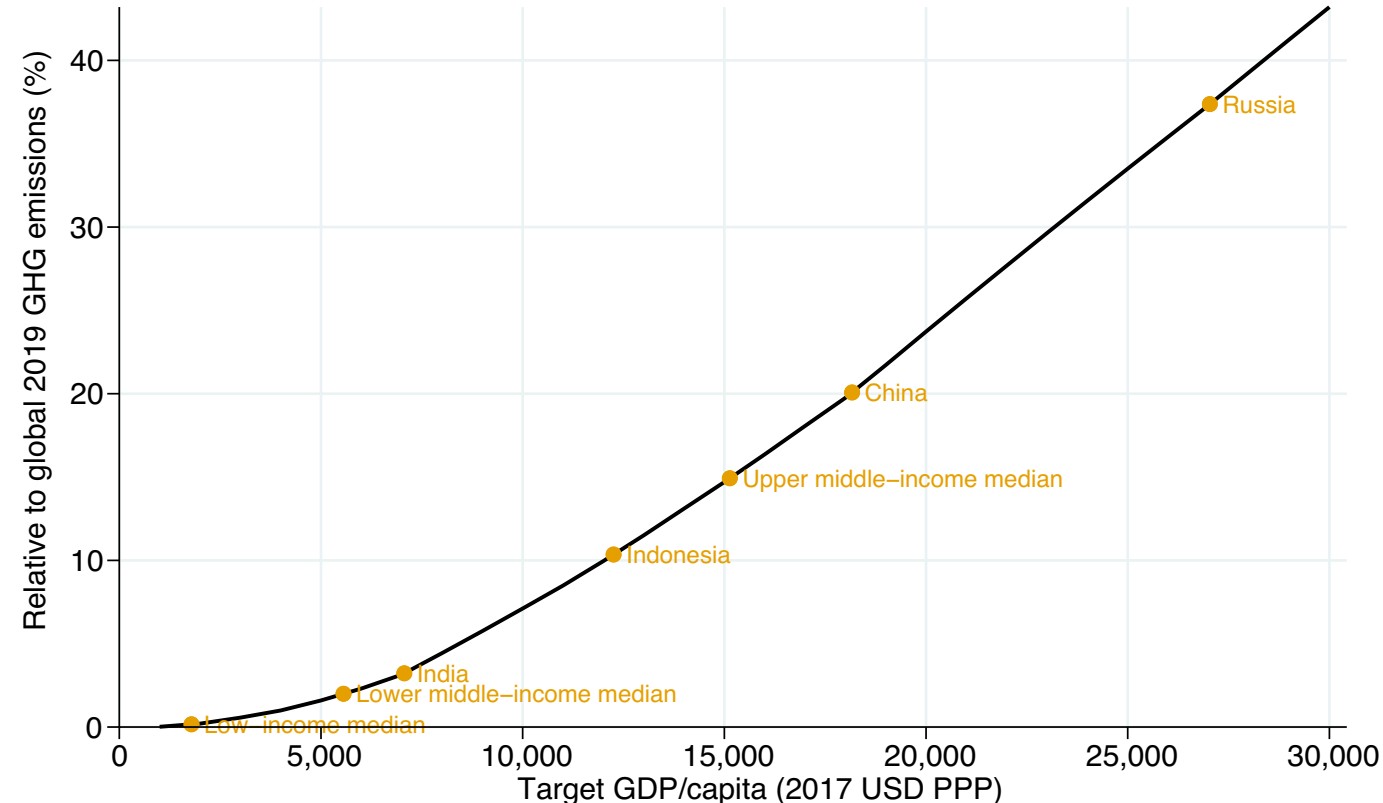

**Extended Data Fig. 7 | CO2e emissions needed to bring all countries to a given GDP *per capita* level by 2050.** The curve shows the estimated CO2e emissions, expressed as percentage of 2019 global emissions, needed for all countries to grow to *per capita* GDP levels shown on the x-axis. Includes all countries with *per capita* GDP under the target *per capita* GDP level. The GDP *per capita* levels in 2019 of selected countries and income-group medians are included for reference.

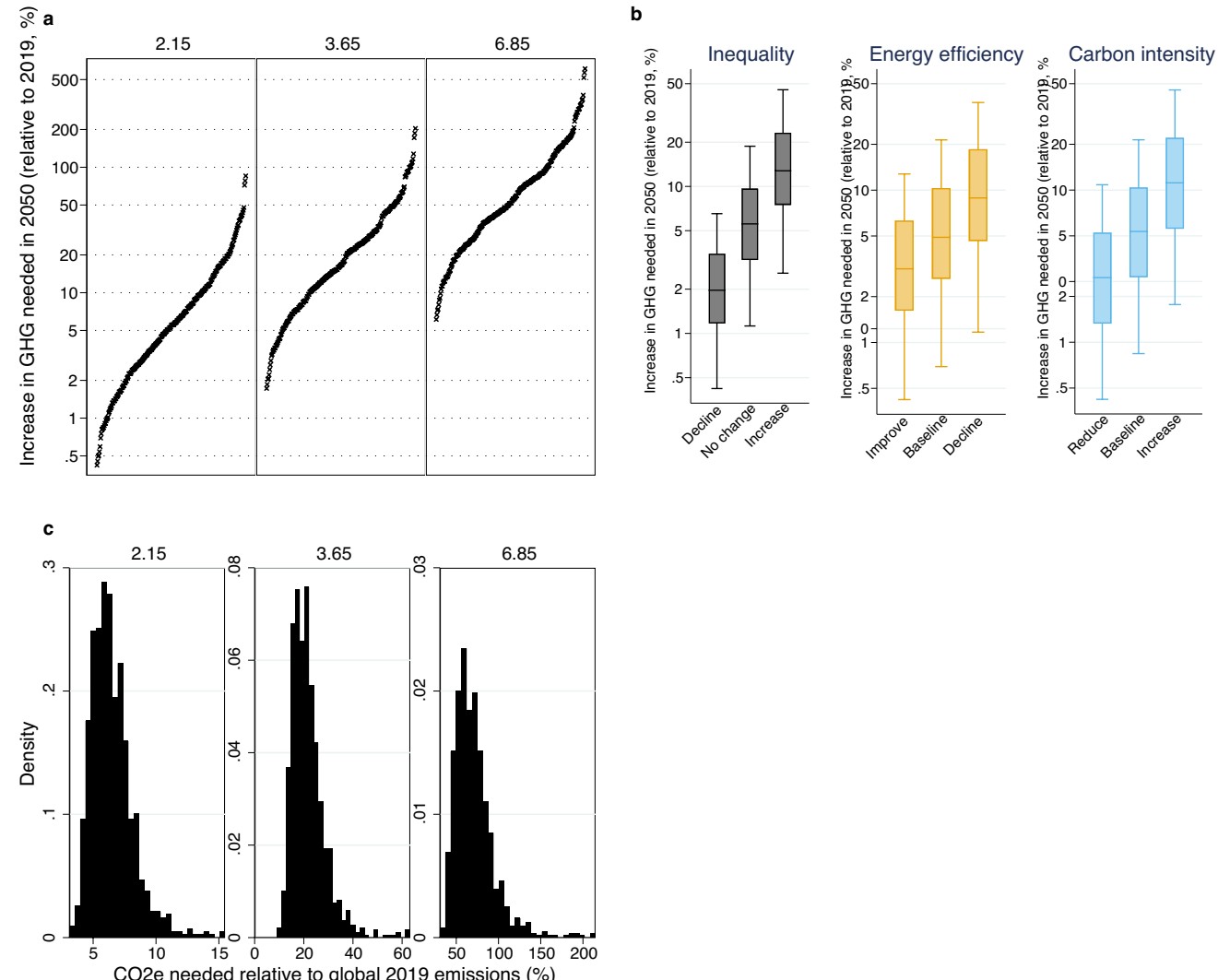

**Extended Data Fig. 8 | CO2e increases of poverty alleviation in 2050 under different scenarios and with uncertainty. a**, plot of CO2e increase for all scenarios at all three poverty lines, **b**, box-plot of scenarios by inequality-change, energy efficiency assumption, and carbon intensity assumption at the $2.15 line. Boxplots show median, 25[th], and 75[th] percentiles. **a**, **b**, the scenarios are described in Extended Data Table 2. **c**, CO2e needed to end extreme poverty and poverty at higher lines when accounting for uncertainty of regressions. Based on 1000 draws of random and fixed effects using the point estimates and standard errors from equations 3, 5, and 7. The point estimate and confidence intervals at the three lines are 4.9% [4.1%–10.8%], 15.3% [13.0%–38.3%], 45.7% [40.4%–129.6%].

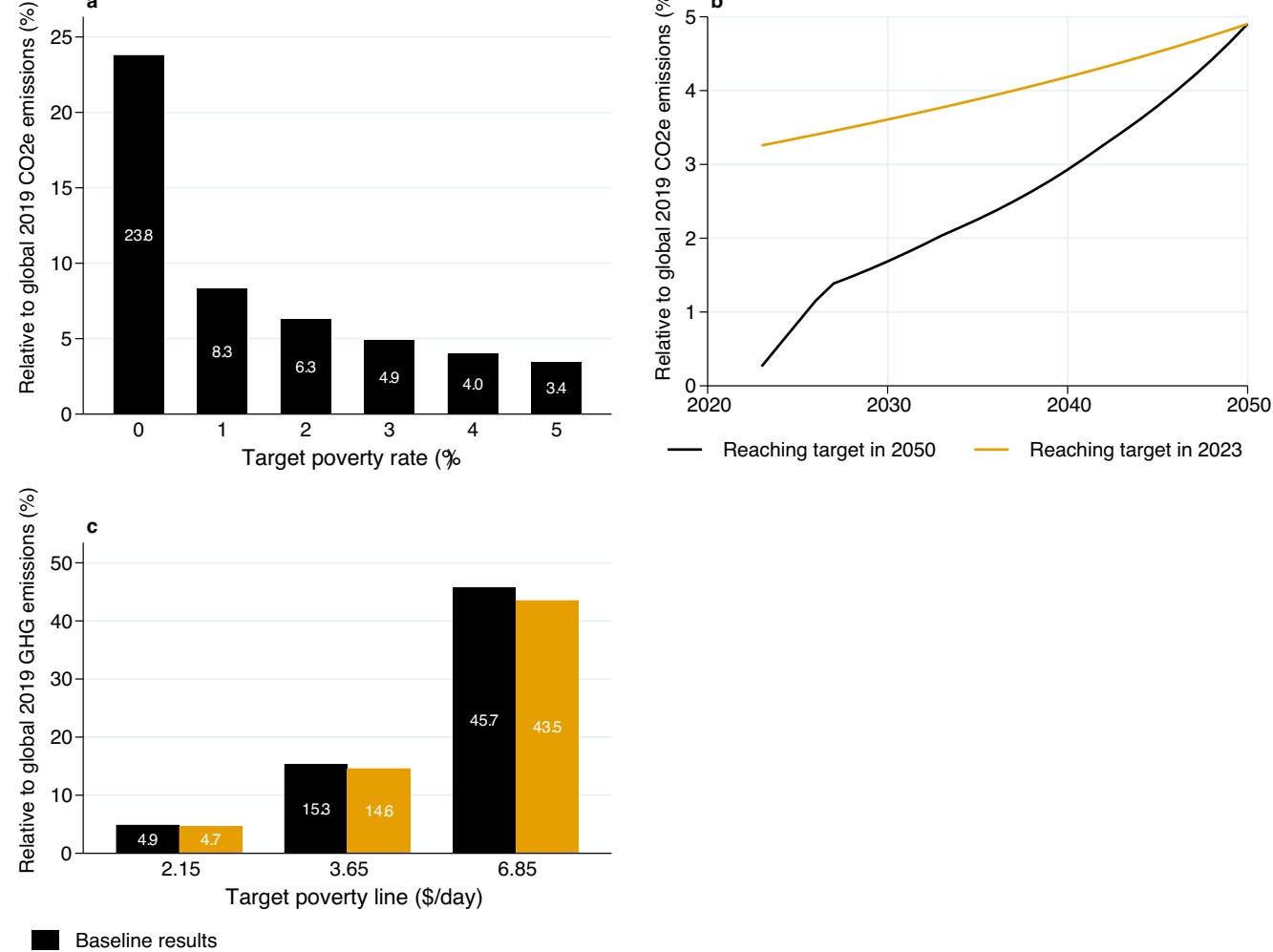

Baseline results

Results when accounting for impact of economic growth on population growth

**Extended Data Fig. 9 | CO2e emissions of poverty alleviation under different assumptions. a**, by target poverty rate in 2050 using the $2.15 line. Results for very low target poverty rates become increasing sensitive to the situation of the poorest households, whose consumption is the hardest to capture and measure and can be linked to idiosyncratic shocks. With the 0% target rate, the results are completely dependent on the consumption of the poorest households in the household survey, making results unreliable. **b**, by target year of reaching 3% using the $2.15 line. The orange line shows the path if all countries reach the GDP *per capita* needed to end extreme poverty in 2023 and then maintain that level onwards to 2050. All the intermediate points on this path are equivalent to the greenhouse gases needed if all countries end extreme poverty by that year. The estimates are increasing over time due to population growth in poor countries. Every year there are more and more people to lift out or maintain out of poverty. This population effect dominates the effect from countries every year being more energy efficient and less carbon intensive. **c**, if accounting for the impact of economic growth on fertility. For the countries not projected to grow sufficiently to end extreme poverty by 2050, and for which the poverty-alleviation scenario mechanically adds growth such that the poverty reduction target is precisely met by 2050, this added economic growth could imply that fertility would fall faster than baseline population projections. Here the decline in population growth associated with this mechanical increase in GDP/capita is estimated, and the population counts are adjusted downwards accordingly.

## Extended Data Table 1 | Regression output

| a | Fixed effect $(\beta)$ | Standard deviation of random effect $(u)$ |
|---|---|---|
| Log GDP *per capita* | 0.701*** (0.038) | 0.291*** (0.033) |
| Constant | -4.170*** (0.345) | 2.730*** (0.320) |

| b | Fixed effect $(\beta)$ | Standard deviation of random effect $(u)$ |
|---|---|---|
| Log GDP *per capita* | 0.998*** (0.038) | 0.332*** (0.045) |
| Year | -0.009*** (0.002) | 0.025*** (0.002) |
| Constant | 18.279*** (3.964) | 49.851*** (4.563) |

| c | Coefficient $(\beta)$ | Standard deviation of random effect $(u)$ |
|---|---|---|
| Log energy *per capita* | 0.689*** (0.030) | 0.303*** (0.048) |
| Year | 0.002 (0.002) | 0.023*** (0.002) |
| Constant | -9.966*** (3.677) | 46.546*** (4.800) |

**a**, Output from a random slope regression predicting consumption *per capita* as a function of GDP *per capita* Note: * = 0.1, ** = 0.05, *** = 0.01. A covariance between the two random effects is estimated as well. Number of countries = 115. Number of observations = 470. Source: Poverty and Inequality Platform, World Bank, and World Development Indicators, World Economic Outlook, and the Maddison Project Database.
**b**, Output from a random slope regression predicting energy *per capita* as a function of GDP *per capita* and time. Note: * = 0.1, ** = 0.05, *** = 0.01. Covariances between the random effects are estimated as well. Number of countries = 193. Number of observations = 1,845. Source: World Development Indicators, World Economic Outlook, the Maddison Project Database, and the U.S Energy Information Administration.
**c**, Output from a random slope regression predicting energy greenhouse gases *per capita* as a function of energy *per capita* and time. Note: * = 0.1, ** = 0.05, *** = 0.01. Covariances between the random effects are estimated as well. Number of countries = 186. Number of observations = 1,774. Source: ClimateWatchData (CAIT) and the U.S Energy Information Administration.

**Extended Data Table 2 | Description of scenarios used**

| Parameter | | Interpretation | Value |
|---|---|---|---|
| Inequality | High | 90th percentile of changes in inequality (increase) | +13% in Gini in 2050 |
| | Baseline | No change | 0% |
| | Low | 10th percentile of changes in inequality (decline) | -17% in Gini in 2050 |
| Passthrough rate of GDP growth to consumption | High | Less growth passes through to consumption | 0.44% increase in consumption for 1% growth in GDP |
| | Baseline | Country-specific estimate of passthrough rate | On average, 0.7% increase in consumption for 1% growth in GDP |
| | Low | More growth passes through to consumption | 0.98% increase in consumption for 1% growth in GDP |
| Energy consumption per GDP | High | 90th percentile of annual improvements in energy efficiency | Deteriorations in energy efficiency of 2.1% per year |
| | Baseline | Country-specific estimate of annual improvements in energy efficiency | On average, improvement in energy efficiency of 0.9% per year |
| | Low | 10th percentile of annual improvements in energy efficiency | Improvement in energy efficiency of 3.2% per year |
| GHG emissions of energy consumption | High | 90th percentile of annual improvement in GHG emissions of energy consumption (increase) | Increase in carbon intensity of energy of 2.5% per year |
| | Baseline | Country-specific estimate of annual improvement in energy intensity of GDP | On average, no change in carbon intensity of energy of 0.1% per year |
| | Low | 10th percentile of annual improvement in energy intensity of GDP (decarbonization) | Reduction in carbon intensity of energy of 2.2% per year |
| Population growth | High | High variant of UN's population projections | Country-specific growth rate |
| | Baseline | World Bank population projections | Country-specific growth rate |
| | Low | Low variant of UN's population projections | Country-specific growth rate |

For each of the five parameters, we combine three realizations, generating 243 (3^5) different scenarios.