## [Peer Review File · Nature]

Manuscript Title: The Climate Implications of Ending Global Poverty

Reviewer Comments & Author Rebuttals

Reviewer Reports on the Initial Version:

Referees' comments:

Referee #1 (Remarks to the Author):

Review of "The Climate Implications of Ending Global Poverty"

This paper takes on an important topic, and both increased economic growth in poor nations and slowing the global impacts of climate change are first order issues of supreme importance. The paper argues that reducing global poverty will not actually stand in the way of climate goals. Historical extrapolations show that pulling up from below to above the poverty lines will not require as much carbon emissions as one might think.

There is a lot to like in this paper, but I do not find the analysis all that convincing in the end of the day. Pulling households above poverty lines is an objective that makes more sense on paper than in practice. In reality, the well-being of poor families in Africa and South Asia does not take a discrete jump up at a poverty line, but continually increases with economic growth. It is valuable in other words to get households up from \$300 to \$600 dollars per year in per capita income. It is also value to get them from \$500 to \$1000 or \$3,500 to \$7,000. The same rate of growth that doubles income from a low level doubles income from a higher one, and I struggle to see the magic of a poverty line as more than a convenient threshold.

The carbon emissions implications of raising living standards from \$3,500 to \$7,000 per capita annual income (say) are sizable and entail a different set of ways to produce emissions. Households in this income range buy cars and even fly in airplanes with some (low but positive) frequency. They buy household durable goods like air conditioners and consume electricity in large quantities, at least when the power is on. It is hard to see how this is an unimportant part of the growth process and that someone one can extrapolate from those right at the poverty line, who have quite different lifestyles.

As economic growth advances, fertility levels tend to fall. The AER paper by Chaterjee and Vogl provide compelling evidence to this effect. How does having fewer bodies around impact emissions? If policy causes faster growth does this reduce fertility in the authors' calculations? The answer seems to be no, since they are using the UN population forecasts.

Over time the efficiencies of greener technologies, including for electricity production, will tend to rise. It is a tall order for the authors to forecast how much they will rise and whether the rate of growth will depend on increases in demand for energy-saving technologies. But this makes it harder for me to buy into the paper's longer run ('Alternative Scenarios') predictions.

Referee #2 (Remarks to the Author):

This paper attempts to quantify the increase in greenhouse gas emissions associated with eradicating poverty. The baseline definition of poverty eradication used by the authors is to bring down the share of the population in extreme poverty to no more than 3%, in every country, by 2050. This benchmark is consistent with the United Nations' interpretation of ending extreme

poverty in the Sustainable Development Goals, with extreme poverty defined as consuming less than \$2.15 per day in 2017 purchasing-power-parity-adjusted USD.

Earlier studies (Bruckner et al., 2022; Hubacek et al., 2017) cited by the authors have attempted a similar exercise. However, these studies have simply calculated the emissions associated with elevating the consumption of the poorest people up to the \$2.15 poverty line. This paper recognizes that poverty eradication is typically driven by broad-based economic growth. It thus tries to calculate the growth rates necessary for each country to eradicate poverty by 2050, and the associated increase in emissions from that growth. Importantly this increase in emissions comes not only from the poor in each country as they rise to the \$2.15 threshold, but also from the non-poor population in these countries, who also consume more due to the overall economic growth and thus emit more greenhouse gases per capita.

The analysis proceeds in 4 steps:

1. The authors determine the consumption growth necessary to meet the target by computing the percentage increase necessary for the 3rd percentile of a country's per-capita consumption distribution to reach \$2.15/day. For instance, the 3rd percentile of Benin's per-capita consumption distribution is currently \$1.33/day; a 62% increase would bring this to \$2.15/day. Importantly, the authors assume that a given percentage increase in a country's average per-capita consumption is achieved by increasing per-capita consumption by the same percentage throughout the distribution (i.e., consumption growth occurs without altering the distribution of consumption).
2. Recognizing that GDP and consumption do not necessarily evolve one-to-one, the authors compute the GDP growth rate necessary for each country to achieve the consumption increase from Step 1. This is accomplished via a regression of consumption-per-capita on GDP per-capita. For instance, for Benin, a percentage increase in GDP per-capita is associated with a 0.68% increase in consumption-per-capita. Thus, to achieve a 62% increase in consumption-per-capita, Benin will require a 91% growth in GDP-per-capita (i.e., $0.62/0.68$) between now and 2050.
3. The authors compute the per-capita energy consumption increase in each country associated with the GDP growth from Step 2. This is accomplished via a regression of per-capita energy consumption on GDP per-capita.
4. Finally, the authors compute the per-capita greenhouse gas emissions increase in each country associated with the energy consumption growth from Step 4. This is accomplished via a regression of per-capita greenhouse gas emissions on energy consumption per-capita. The per capita increase in emissions is converted to a total increase by multiplying by the country's projected population, and lastly summing across all countries.

The paper finds that growth necessary to achieve the 3%/\$2.15/2050 target will result in an annual emissions increase of 2.37 GtCO_{2e} in 2050 (equivalent to 4.9% of 2019 global emissions). They claim that this finding is somewhat larger than those of earlier studies but of similar order of magnitude- the earlier studies find an emissions increase of 1%-3% when only considering the emissions associated with elevating the consumption of the poorest people up to the poverty line. Even though this paper accounts for the broader growth and associated emissions needed to reach the target, it still finds that the additional emissions needed for reaching the 3%/\$2.15/2050 target are relatively small.

My main concern with this paper is how it positions its findings vis-à-vis those of previous studies. The previous studies by Bruckner et al. (2022) and Hubacek et al. (2017) calculate the emissions increase associated total elimination of extreme poverty globally, and they find the increase to be in the range of 1%-3%. This paper focuses on a different, less stringent goal, i.e. reducing the share of people in extreme poverty to no more than 3% in each country, and calculates a 4.9% emissions increase associated with reaching that goal. However, when considering the same goal

as the previous studies (i.e., 0% extreme poverty), the authors find a much larger emissions increase of 23.8% (see Extended Data Figure 5, Panel A). Thus I feel this study, in trying to account for the broad-based growth necessary for poverty alleviation, comes to a qualitatively different conclusion than previous work. As indicated in Extended Data Figure 5, Panel A, the “last mile” of poverty eradication (i.e., to get below 1% extreme poverty in each country) entails a considerable increase in emissions, due to the high growth rates necessary to lift the consumption levels of the very poorest up to the \$2.15 threshold. The authors should bring out this nuance when explaining and discussing the results.

The methodology used in this paper is essentially that of reduced-form econometrics as opposed to a structural model of the macroeconomy. This approach has the advantage of being transparent, but it omits certain general equilibrium aspects. For instance, it is possible that the overall economic growth in poor countries necessary to lift consumption of the poorest also results in higher emissions in rich countries due to international economic linkages. If so, this paper could be underestimating the emissions increase associated with poverty alleviation. While I do not think it is necessary to adopt a structural approach, at the minimum, a discussion of some of these limitations of the reduced form approach and implications for the results would be useful.

In its main results, this paper assumes distribution-neutrality of consumption growth, i.e. consumption grows at the same rate throughout the distribution, thus keeping inequality constant. However, recent work has questioned whether distribution-neutrality is a reasonable assumption, particularly noting that the effect of growth on inequality can vary greatly across countries and across time. (See for example Ravallion, 2022: “We no longer live in a world where ... distribution-neutrality ... can be considered plausible.”) In principle, one could use data to shed light on this question. Specifically, regression equation (3) could be estimated with the dependent variable being the percent of country c 's population below the \$2.15 poverty line in year y . The results of this regression could be directly used to calculate the GDP growth necessary for each country to reach 3%. In practice, I suppose this approach was not implemented due to the poverty rate data being limited in its coverage across countries/years (<https://data.worldbank.org/indicator/SI.POV.DDAY>). Because of these data limitations, it is understandable that a regression of the poverty rate on GDP is not adopted in the primary approach. However, it would still be useful as a robustness check to run such a regression on selected countries with relatively good data coverage, and for comparison purposes, calculate in this way the GDP growth necessary for these countries to reach 3%.

Other points:

Regression equations (5) and (7) include country-specific linear time trends, but regression equation (3) does not. What is the reason for this distinction? Do the estimates in (3) change substantially if the country-specific trend is included?

While the authors report the standard errors of the regression coefficients in Extended Data Tables 2 and 3, this uncertainty is not propagated through to the subsequent calculations (expressed in equations 4,6,8,9,10). The authors should resample from the regression estimates, carry forward the uncertainty, and report confidence intervals or standard errors for the final results.

References:

Bruckner, B., Hubacek, K., Shan, Y., Zhong, H. and Feng, K., 2022. Impacts of poverty alleviation on national and global carbon emissions. *Nature Sustainability*, 5(4), pp.311-320.

Hubacek, K., Baiocchi, G., Feng, K. and Patwardhan, A., 2017. Poverty eradication in a carbon constrained world. *Nature communications*, 8(1), p.912.

Ravallion, M., 2022. Growth Elasticities of Poverty Reduction (No. w30401). National Bureau of

Economic Research.

Referee #3 (Remarks to the Author):

This is a review for the manuscript, "The Climate Implications of Ending Global Poverty," by Wollberg et al. In this analysis, the authors extrapolate relationships of carbon intensity of economic activity to estimate the greenhouse gas emissions implications of eradicating extreme poverty, poverty and bringing all households up above the lower-middle income poverty line. The overarching finding presents an important, policy-relevant conclusion that the global eradication of extreme poverty is likely to have negligible effects on greenhouse gas emissions and global capacity to meet climate policy goals. On the other hand, the authors contend that meeting more ambitious poverty-reduction goals presents a bigger tradeoff between poverty reduction and climate change related goals.

The analysis presented is not particularly methodologically innovative - it presents mostly a series of linear fits/extrapolations using widely applied datasets, however it addresses an important topic and presents its findings in a way that is very accessible to a general audience. The trope that the paper aims to debunk, that alleviation of extreme poverty is somehow at odds with climate risk management, is an extremely damaging one in international policy debates and for that reason I think publication in Nature would be valuable. However, there are some shortcomings of the analysis that need to be substantively addressed first.

Methodologically, and considering robustness of the conclusions, I'd consider the findings about extreme poverty more defensible than those for the higher poverty thresholds examined. This is because the manuscript essentially presents a marginal analysis, requiring assumptions of no dynamic economic feedbacks to the large increases in demand and consumption in some of the scenarios, an assumption which I'm not sure holds beyond the extreme poverty case which requires a relatively small perturbation to the current global consumption. A 50% increase in energy consumption globally (the increase associated with the lower-middle income threshold) would have impacts on consumption and prices elsewhere — once the analysis becomes non-marginal, it requires a more sophisticated (dynamic) economic model to really be credible

For "less inequality" scenario, I also wonder why the authors chose to just implement this in poor countries— it would be interesting, and in some ways more meaningful, to implement this globally not just in poor countries to see its effect— this might even give a global reduction in emissions.

Smaller comments:

- In a couple of places in the manuscript, including in the abstract, the authors write that "the need to eradicate extreme poverty cannot be used as a justification for reducing the world's climate ambitions". I would suggest stating the converse is true as well, that, nor should the need for climate action be used as an excuse to ignore the urgency of investments in poverty reduction.
- Summary paragraph say that "poverty reduction comes about because of economic growth"- might say 'has historically come about'
- 3rd para of p2 says "90% of historical poverty alleviation driven by economic growth" - implying domestic— please confirm
- Figure 1: these panels seem duplicative — can you just label what's in panel A better, then panel B is not required
- Bottom of page 4, setting up the counterfactual states that you assume there "is no growth in per capita GDP in poor countries" — this seems unlikely (and inconsistent with historical trends) — I don't think it's a fair assumption and it will inflate the significance of the results
- Last paragraph p.7, regarding my earlier comment, why just poor countries??
- First paragraph p.8, would say "is unlikely" instead of "cannot be expected"
- Final paragraph, p.8: again when saying "and the need to eradicate extreme poverty cannot be

used as a justification for reducing the world's climate ambitions" would specify 'and vice versa' too

- Final sentence, "Our analysis faces important limitations." This is an extremely abrupt and strange concluding sentence. I agree there are important limitations, a number of which I touched up on in my general comments. You need to elaborate substantially here.

Author Rebuttals to Initial Comments:

Editor

Your manuscript, "The Climate Implications of Ending Global Poverty", has now been seen by 3 referees, whose comments are attached below. In the light of their advice we have decided that we cannot offer to publish the manuscript in Nature.

While the referees find your work of some interest, they raise serious concerns about the appropriateness of the technical approach, the underlying assumptions and the real-world relevance. We feel that these reservations are sufficiently important as to preclude publication of the present study in Nature, and we are accordingly closing the file.

Having said this, should future experimental data and theoretical analysis allow you to address these concerns by sufficiently justifying your assumptions with an improved approach, we would be happy to look at a revised manuscript (unless, of course, something similar has by then been accepted at Nature or appeared elsewhere). At such a time, you can request that the manuscript file be reopened for resubmission by simply sending me an email to that effect (stating manuscript number). In the case of eventual publication, the received date would be that of the revised paper. I should stress, however, that we would be reluctant to trouble our referees again unless we thought that their comments and any editorial issues had been addressed in full, and we would understand if you prefer instead to now pursue publication of the work elsewhere.

I am sorry that we cannot respond more positively, and we hope that you find our referees' comments helpful.

Thank you for the very helpful feedback from you and the three referees. We have had a very careful look at the comments we received and think that we have been able to respond to their concerns and justify our assumptions with improved arguments, new data, and new theoretical analysis. Most important was to clarify our objective, which is not to forecast future poverty, but to use scenarios to understand the relationship between poverty and emissions, under various assumptions for inequality, carbon and energy intensity. Concretely, we have taken the following steps, which we outline in much more detail in the itemized responses:

- *Argue more clearly why using a poverty line is sensible and include new data and analysis on what would happen if we instead targeted a certain GDP/capita level across all countries (in response to referee 1).*
- *Include new analysis accounting for the impact of growth on fertility (in response to referee 1). Our results do not change much.*
- *Benchmark our alternative scenarios with SSP scenarios to increase their credibility (in response to referee 1). Our scenarios are well aligned with SSP1-26.*
- *Position our findings better vis-a-vis existing studies (in response to referee 2).*
- *Explain better our use the distribution-neutral projection (in response to referee 2).*
- *Test the inclusion of time-effects in the GDP-to-consumption regression (in response to referee 2)*
- *With new data and analysis, propagate standard errors from our random-slope regression to get bounds on our main results (in response to referee 2).*

- *Better argue for our methodological approach while adding more discussion on its limitations (in response to referee 3).*
- *Discuss why decreasing inequality of non-poor countries would not matter for the results (in response to referee 3).*

We think the new version of the paper is much improved and hope you agree. Needless to say, we would be happy to address any further concerns as needed.

Referee #1 (Remarks to the Author):

Review of “The Climate Implications of Ending Global Poverty”

This paper takes on an important topic, and both increased economic growth in poor nations and slowing the global impacts of climate change are first order issues of supreme importance. The paper argues that reducing global poverty will not actually stand in the way of climate goals. Historical extrapolations show that pulling up from below to above the poverty lines will not require as much carbon emissions as one might think.

There is a lot to like in this paper, but I do not find the analysis all that convincing in the end of the day. Pulling households above poverty lines is an objective that makes more sense on paper than in practice. In reality, the well-being of poor families in Africa and South Asia does not take a discrete jump up at a poverty line, but continually increases with economic growth. It is valuable in other words to get households up from \$300 to \$600 dollars per year in per capita income. It is also value to get them from \$500 to \$1000 or \$3,500 to \$7,000. The same rate of growth that doubles income from a low-level doubles income from a higher one, and I struggle to see the magic of a poverty line as more than a convenient threshold.

The carbon emissions implications of raising living standards from \$3,500 to \$7,000 per capita annual income (say) are sizable and entail a different set of ways to produce emissions. Households in this income range buy cars and even fly in airplanes with some (low but positive) frequency. They buy household durable goods like air conditioners and consume electricity in large quantities, at least when the power is on. It is hard to see how this is an unimportant part of the growth process and that someone one can extrapolate from those right at the poverty line, who have quite different lifestyles.

This concern is about the use of poverty lines and targets in our paper. We would like to make two points in response. First, we contend that poverty lines are the most relevant and widely used policy metric when it comes to poverty, and it is thus the right metric to use to position our analysis in the most relevant way possible. Second, our approach is flexible and general enough to analyze the emissions implications of raising living standards in terms of GDP per capita in the same way as and together with poverty lines, which we do in the revised paper. We explain each point in more detail below.

On the first point: We understand the reviewer’s concerns with poverty lines, and to be sure, nothing magical happens when a household moves from just below to just above a poverty line. However, poverty lines are the most widely used and therefore relevant policy metric when it comes to poverty. In many countries of the world, there are social protection programs that are targeted at people living in poverty, that is, below a specified national poverty line. The international poverty lines are designed to reflect those national poverty lines. Poverty eradication is enshrined in the first Sustainable Development Goal, and the

SDG uses the same extreme poverty line we employ. The extreme poverty line is designed to mark a minimum standard of living or cost of basic needs, so this threshold is arguably more meaningful than just any threshold, which is why it is the target of SDG 1. In all, poverty eradication defined relative to a poverty line is widely employed and relevant in both national policy making and the international policy dialogue, and as a result it is also a widely studied concept in the academic literature. In light of this, for our study to be able to meaningfully and impactfully speak to the scientific and policy debates around poverty, growth, climate change, we contend it is necessary and correct to employ poverty lines.

On the second point: In our approach, per capita GDP growth and poverty eradication are directly linked, a key strength of the approach that sets our study apart from previous studies. Incidentally, this is how poverty alleviation comes about most of the time: growth in per capita GDP that benefits the entire population, including the poorest. We can therefore analyze poverty eradication and per capita GDP growth in the same model. For example, in the figure below, which we now include in the paper, we show the greenhouse gas emissions needed to bring all countries to any given GDP per capita level from \$1,000 to \$30,000. We have added some reference GDP per capita levels to make the results more relatable.

Figure 1: CO2e emissions needed to bring all countries to a given GDP/capita level by 2050

Evidently, bringing all countries to the typical GDP/capita of upper middle-income countries would take drastically fewer emissions than to end poverty at the typical upper middle-income poverty line. The reason is that even at the GDP/capita of upper-middle income countries, many households have incomes far below that per capita level and live in poverty. We have added a discussion on this to the paper. It reinforces our point, we believe, that reducing inequality allows one to end poverty at lower costs to the environment.

As economic growth advances, fertility levels tend to fall. The AER paper by Chatterjee and Vogl provide compelling evidence to this effect. How does having fewer bodies around impact emissions? If policy causes faster growth does this reduce fertility in the authors’ calculations? The answer seems to be no, since they are using the UN population forecasts.

This is a great point, which we have accounted for. As a baseline, we use UN population projections. For the countries that are projected to have sufficient economic growth to end poverty by 2050, like India, the UN's population projections are presumably already accounting for the impact of economic development in their projections. However, for the countries that are not projected to grow sufficiently to end poverty by 2050, like DRC, and for which we mechanically add growth such that the poverty target is precisely met by 2050, this added economic growth would likely imply that fertility would fall faster than UN projects.

Our modelling framework allows us to account for this by estimating the relationship between per capita GDP and fertility/population growth and modelling future population growth based on this estimated relationship. More specifically, if a country is 'forced' to grow to a per capita GDP of \$15,000 even though current growth projections suggest it would only reach \$10,000 by 2050, we can estimate the decline in the population growth rate associated with this increase in GDP/capita, and adjust downwards the UN population growth rates by this amount.

Doing so pushes down our main results a little bit, as shown in Figure 2 below, but not enough to change our main conclusion. We have now added this figure to the paper and discuss the impact of growth on fertility.

Figure 2: Results if accounting for the impact of economic growth on fertility

Over time the efficiencies of greener technologies, including for electricity production, will tend to rise. It is a tall order for the authors to forecast how much they will rise and whether the rate of growth will depend on increases in demand for energy-saving technologies. But this makes it harder for me to buy into the paper's longer run ('Alternative Scenarios') predictions.

We agree it is hard to forecast how much efficiencies of greener technologies will rise until 2050. Partially for that reason, we are not attempting to do any forecasts in the paper, rather we are modelling what would happen if historical patterns continue in the future. We have significantly edited the paper to make this point stand out. Ultimately, policy choices will determine (at least to an extent) what the future of energy efficiency will look like.

Obviously, the usefulness of this approach hinges on the relevance of benchmarking with historical patterns, and for the alternative scenarios section, best-performing historical patterns. In the revised version, we have tried to contextualize our alternative scenarios better by benchmarking them with the Shared Socioeconomic Pathways.

In the baseline scenario, we extrapolate from the rate of improvement in energy efficiency based on data from the last 10 years. In the high-efficiency scenarios we present in the paper, we instead model energy efficiency to follow the path of what we have termed the historical ‘best performers’ – that is, the 90th percentile of improvements in energy and carbon efficiency of the past 10 years. This is an optimistic but plausible pathway as it has been observed in the past. In this high-efficiency scenario, the rate of improvement in energy efficiency is 2.2% per year and in carbon intensity it is 3.2% per year. For reference, in the Shared Socioeconomic Pathway (SSP) 1-19 which is compatible with keeping to a 1.5C warming target, improvement in energy efficiency is 3.8% per year, and 5.5% for carbon intensity. In SSP1-26, compatible with a 2C warming target, these rates are 3.4% and 2.4%, respectively. The SSPs are based on a large set of parameters about possible futures and are widely used as reference points in the academic and policy literatures. That our high-efficiency scenario is in a comparable range gives us some confidence in the plausibility of this scenario. We had added a short discussion of this to the paper.

Referee #2 (Remarks to the Author):

This paper attempts to quantify the increase in greenhouse gas emissions associated with eradicating poverty. The baseline definition of poverty eradication used by the authors is to bring down the share of the population in extreme poverty to no more than 3%, in every country, by 2050. This benchmark is consistent with the United Nations’ interpretation of ending extreme poverty in the Sustainable Development Goals, with extreme poverty defined as consuming less than \$2.15 per day in 2017 purchasing-power-parity-adjusted USD.

Earlier studies (Bruckner et al., 2022; Hubacek et al., 2017) cited by the authors have attempted a similar exercise. However, these studies have simply calculated the emissions associated with elevating the consumption of the poorest people up to the \$2.15 poverty line. This paper recognizes that poverty eradication is typically driven by broad-based economic growth. It thus tries to calculate the growth rates necessary for each country to eradicate poverty by 2050, and the associated increase in emissions from that growth. Importantly this increase in emissions comes not only from the poor in each country as they rise to the \$2.15 threshold, but also from the non-poor population in these countries, who also consume more due to the overall economic growth and thus emit more greenhouse gases per capita.

The analysis proceeds in 4 steps:

1. The authors determine the consumption growth necessary to meet the target by computing the percentage increase necessary for the 3rd percentile of a country’s per-capita consumption distribution to reach \$2.15/day. For instance, the 3rd percentile of Benin’s per-capita consumption distribution is currently \$1.33/day; a 62% increase would bring this to \$2.15/day. Importantly, the authors assume that a given percentage increase in a country’s average per-capita consumption is achieved by increasing per-capita consumption by the same percentage throughout the distribution (i.e., consumption growth occurs without altering the distribution of consumption).

2. Recognizing that GDP and consumption do not necessarily evolve one-to-one, the authors compute the GDP growth rate necessary for each country to achieve the consumption increase from Step 1. This is accomplished via a regression of consumption-per-capita on GDP per-capita. For instance, for Benin, a percentage increase in GDP per-capita is associated with a 0.68% increase in consumption-per-capita. Thus, to achieve a 62% increase in consumption-per-capita, Benin will require a 91% growth in GDP-per-capita (i.e., $0.62/0.68$) between now and 2050.

3. The authors compute the per-capita energy consumption increase in each country associated with the GDP growth from Step 2. This is accomplished via a regression of per-capita energy consumption on GDP per-capita.

4. Finally, the authors compute the per-capita greenhouse gas emissions increase in each country associated with the energy consumption growth from Step 4. This is accomplished via a regression of per-capita greenhouse gas emissions on energy consumption per-capita. The per capita increase in emissions is converted to a total increase by multiplying by the country's projected population, and lastly summing across all countries.

The paper finds that growth necessary to achieve the 3%/2.15/2050 target will result in an annual emissions increase of 2.37 GtCO₂e in 2050 (equivalent to 4.9% of 2019 global emissions). They claim that this finding is somewhat larger than those of earlier studies but of similar order of magnitude- the earlier studies find an emissions increase of 1%-3% when only considering the emissions associated with elevating the consumption of the poorest people up to the poverty line. Even though this paper accounts for the broader growth and associated emissions needed to reach the target, it still finds that the additional emissions needed for reaching the 3%/2.15/2050 target are relatively small.

My main concern with this paper is how it positions its findings vis-à-vis those of previous studies. The previous studies by Bruckner et al. (2022) and Hubacek et al. (2017) calculate the emissions increase associated total elimination of extreme poverty globally, and they find the increase to be in the range of 1%-3%. This paper focuses on a different, less stringent goal, i.e. reducing the share of people in extreme poverty to no more than 3% in each country, and calculates a 4.9% emissions increase associated with reaching that goal. However, when considering the same goal as the previous studies (i.e., 0% extreme poverty), the authors find a much larger emissions increase of 23.8% (see Extended Data Figure 5, Panel A). Thus I feel this study, in trying to account for the broad-based growth necessary for poverty alleviation, comes to a qualitatively different conclusion than previous work. As indicated in Extended Data Figure 5, Panel A, the "last mile" of poverty eradication (i.e., to get below 1% extreme poverty in each country) entails a considerable increase in emissions, due to the high growth rates necessary to lift the consumption levels of the very poorest up to the \$2.15 threshold. The authors should bring out this nuance when explaining and discussing the results.

We understand this concern and have repositioned our paper to bring better clarity on how it relates to the prior studies. In particular, we have done the following:

- *Explicitly make it clear that our focus on a 3% target is different from the prior studies, which focused on 0%.*
- *Make it clear that getting to 0% is not meaningful in our framework because (1) there will always be some transitory poverty, and (2) measurement error will lead some households to have an improbable low consumption on which one cannot live. For that reason, we have also removed the*

results with a 0% target in (now) Extended Data Figure 12 – it probably lends itself to misunderstandings and is itself a biased estimate.)

- *Explicitly mention that our results increase notably the closer we get towards a 0% target, and hence that the last mile of poverty reduction might be more carbon intensive.*
- *Note that if the goal is to bring poverty close to 0%, say to 1% in all countries, then we find more significant emissions, which exceed estimates of previous studies by even more than our headline results.*

The methodology used in this paper is essentially that of reduced-form econometrics as opposed to a structural model of the macroeconomy. This approach has the advantage of being transparent, but it omits certain general equilibrium aspects. For instance, it is possible that the overall economic growth in poor countries necessary to lift consumption of the poorest also results in higher emissions in rich countries due to international economic linkages. If so, this paper could be underestimating the emissions increase associated with poverty alleviation. While I do not think it is necessary to adopt a structural approach, at the minimum, a discussion of some of these limitations of the reduced form approach and implications for the results would be useful.

This is a great point, which we now mention in the end when discussing our limitations.

In its main results, this paper assumes distribution-neutrality of consumption growth, i.e. consumption grows at the same rate throughout the distribution, thus keeping inequality constant. However, recent work has questioned whether distribution-neutrality is a reasonable assumption, particularly noting that the effect of growth on inequality can vary greatly across countries and across time. (See for example Ravallion, 2022: “We no longer live in a world where ... distribution-neutrality ... can be considered plausible.”) In principle, one could use data to shed light on this question. Specifically, regression equation (3) could be estimated with the dependent variable being the percent of country c 's population below the \$2.15 poverty line in year y . The results of this regression could be directly used to calculate the GDP growth necessary for each country to reach 3%. In practice, I suppose this approach was not implemented due to the poverty rate data being limited in its coverage across countries/years (<https://data.worldbank.org/indicator/SI.POV.DDAY>). Because of these data limitations, it is understandable that a regression of the poverty rate on GDP is not adopted in the primary approach. However, it would still be useful as a robustness check to run such a regression on selected countries with relatively good data coverage, and for comparison purposes, calculate in this way the GDP growth necessary for these countries to reach 3%.

These are scenarios to understand the implications of eradicating poverty in different ways (with distribution-neutral growth, with historical energy and carbon patterns, etc.), but those are not forecasts. They cannot be forecast because the future depends on policies and future policies are deeply uncertain. Yet we maintain that assuming distribution-neutrality is an appropriate approximation for the following reasons (which we would have brought about more clearly in the paper):

- *Recent evidence from Bergstrom (2022, <https://doi.org/10.1093/wber/lhab026>) shows that 90% of changes to poverty have come from distribution-neutral growth rather than changes to inequality.*
- *This is consistent with the bulk of Ravallion (2022) which states that “one of the stylized facts to emerge by the early 2000’s is that observed growth processes tend to be distribution-neutral on average (Ravallion 2001; Dollar and Kraay 2002; Ferreira and Ravallion 2009).” Ravallion’s*

conclusion, which the referee cites, seems to be based on rather weak evidence suggesting that inequality in developing countries has been increasing. A more systematic analysis does not find this to be the case (Poverty and Shared Prosperity Report 2022).

- *Running a regression with poverty rates on the left-hand side would ignore the rich distributional information we possess, such as ignoring whether countries have a lot of mass close to the poverty line.*

All this said, it is of course possible that distribution-neutrality will not hold on average in the future, for example if extreme weather events hit the poorest in each country hardest, leading (in the absence of policy responses) to higher inequality. We have added a discussion on this.

Other points:

Regression equations (5) and (7) include country-specific linear time trends, but regression equation (3) does not. What is the reason for this distinction? Do the estimates in (3) change substantially if the country-specific trend is included?

This is a great question which we regret not having discussed in the paper. We omitted the time trend in equation (3) for three reasons: (a) We had no theoretical prior to suggest a country-specific linear time trend. It would suggest that conditional on a given level of GDP/capita, every year countries continuously increase (or decrease) mean consumption without any change to income (or equivalently that the savings rate constantly increases (or decreases)). (b) If we do include year in the regression, the fixed effect is highly insignificant. (c) Due to the poverty data not being annual in most countries, we have much less power to include country-specific linear time trends. All this said, we have tried to add it to our model. Our main results change a bit, see Figure 3. This is mainly driven by odd country-level results, and notably India. For example, since the country-specific time trend for India is about 0.01 (which is among the more extreme values), with this specification, India's consumption vector grows 1% per year even in the absence of GDP/growth. This means that the savings rate continuously falls. Though India is estimated to have an extreme poverty rate of 9.4% in 2022, equivalent to more than 100 million people in extreme poverty, with this specification, India reaches the target of 3% by 2050 without any GDP growth and hence without any added greenhouse gas emissions. We think the oddity of these results speak against including them as a robustness check but would be happy to follow guidance on this. We have added a paragraph discussing the choice of not including country-specific linear time trends.

Figure 3: Results when using a country-specific linear time-trend in the consumption-GDP regression

While the authors report the standard errors of the regression coefficients in Extended Data Tables 2 and 3, this uncertainty is not propagated through to the subsequent calculations (expressed in equations 4,6,8,9,10). The authors should resample from the regression estimates, carry forward the uncertainty, and report confidence intervals or standard errors for the final results.

That is a great point. We have now done so, and report our results from 1000 resamples in the paper. Our confidence intervals for our three main results become 4.9% [3.9%-8.6%], 15.3% [11.7%, 29.8%], 45.7% [36.7%, 94.5%]. We think that results are qualitatively unchanged --- ending extreme poverty with historical rates of improvements in carbon and energy efficiency does not require a massive addition to global emissions, while ending poverty at higher lines does, especially at the upper middle-income line.

We now include these confidence intervals in the paper, but it is worth noting that they do not account for all conceivable uncertainty; there is also uncertainty related to the projection of the welfare distributions, the imputations of data for countries without primary data, the population numbers, etc. It is also worth noting that the uncertainty reported below would go down if we lose data for one country (as a random coefficient would no longer be possible to estimate), which is a bit counterintuitive. Likewise, uncertainty would go down if we winsorize the random coefficients more.

Figure 4: Results from 1000 draws of random coefficients

Referee #3 (Remarks to the Author):

This is a review for the manuscript, “The Climate Implications of Ending Global Poverty,” by Wollberg et al. In this analysis, the authors extrapolate relationships of carbon intensity of economic activity to estimate the greenhouse gas emissions implications of eradicating extreme poverty, poverty and bringing all households up above the lower-middle income poverty line. The overarching finding presents an important, policy-relevant conclusion that the global eradication of extreme poverty is likely to have negligible effects on greenhouse gas emissions and global capacity to meet climate policy goals. On the other hand, the authors contend that meeting more ambitious poverty-reduction goals presents a bigger tradeoff between poverty reduction and climate change related goals.

The analysis presented is not particularly methodologically innovative - it presents mostly a series of linear fits/extrapolations using widely applied datasets, however it addresses an important topic and presents its findings in a way that is very accessible to a general audience. The trope that the paper aims to debunk, that alleviation of extreme poverty is somehow at odds with climate risk management, is an extremely damaging one in international policy debates and for that reason I think publication in Nature would be valuable. However, there are some shortcomings of the analysis that need to be substantively addressed first.

Methodologically, and considering robustness of the conclusions, I'd consider the findings about extreme poverty more defensible than those for the higher poverty thresholds examined. This is because the manuscript essentially presents a marginal analysis, requiring assumptions of no dynamic economic feedbacks to the large increases in demand and consumption in some of the scenarios, an assumption which I'm not sure holds beyond the extreme poverty case which requires a relatively small perturbation to the current global consumption. A 50% increase in energy consumption globally (the increase associated with the lower-middle income threshold) would have impacts on consumption and prices elsewhere — once the analysis becomes non-marginal, it requires a more sophisticated (dynamic) economic model to really be credible.

We certainly understand this point of view. In part because of it, we are not attempting to do any forecasts in the paper, rather we are modelling what would happen if historical patterns continue in the future. We have significantly edited the paper to make this point stand out. Obviously, the usefulness of this approach hinges on the relevance of benchmarking with historical patterns. This is particularly challenging when we need to model a world where poverty ends at the middle-income poverty lines. This is not only because of the complex dynamic relationships between consumption, prices, energy, and more, but also because such a world, unfortunately, is fundamentally out of reach. What does a world look like in which everyone in countries like Yemen, Somalia, and Haiti have consumption levels of at least of middle-income standards? We think even a complicated dynamic model will be unable to answer this question with accuracy and have favored a tractable model where the assumptions are laid out clearly. Yet to account for this very important comment, we have done the following in the revised version of the paper:

- *Benchmark our model against historical patterns. Notably, from 2001-2019, global energy consumption increased by 2.2% per year. In our scenario where global poverty will be ended at the \$6.85 line, global energy consumption will increase by 2.6% per year from 2023-2050. Hence, the total increase in energy and consumption is not of completely different magnitudes.*
- *Tackle one of the most obvious dynamic relationship through our current set-up by modelling the dynamic effect of growth on fertility.*
- *Discuss the impact of some of the dynamic relationships which are difficult to account for in any model, such as the impact of one country's growth on that of all other countries.*
- *We have removed any reference to results if poverty need to be eradicated at \$15 line, as this would require even more growth in production and energy.*

For “less inequality” scenario, I also wonder why the authors chose to just implement this in poor countries— it would be interesting, and in some ways more meaningful, to implement this globally not just in poor countries to see its effect— this might even give a global reduction in emissions.

We implemented reduced inequality in poor countries since we only look at the emissions from growth in poor countries (i.e. countries with more than 3% of the population below the poverty line). These are the only countries that need to grow to end poverty. Note that a reduction in inequality in non-poor countries without a change in mean consumption would not impact emissions under the assumption that the elasticity of emissions to consumption is unity, which we think there is not sufficient evidence to reject.

Smaller comments:

- In a couple of places in the manuscript, including in the abstract, the authors write that “the need to eradicate extreme poverty cannot be used as a justification for reducing the world’s climate ambitions”. I would suggest stating the converse is true as well, that, nor should the need for climate action be used as an excuse to ignore the urgency of investments in poverty reduction

Good point, we have now added this

.- Summary paragraph say that “poverty reduction comes about because of economic growth”- might say ‘has historically come about’

Now edited.

- 3rd para of p2 says “90% of historical poverty alleviation driven by economic growth” - implying domestic— please confirm

That is right, we mean domestic economic growth.

- Figure 1: these panels seem duplicative — can you just label what’s in panel A better, then panel B is not required

Good point, we have now done so.

- Bottom of page 4, setting up the counterfactual states that you assume there “is no growth in per capita GDP in poor countries “ — this seems unlikely (and inconsistent with historical trends) — I don’t think it’s is a fair assumption and it will inflate the significance of the results

We certainly agree that it is unlikely and inconsistent with historical trends that poor countries will not grow. Our intention with the counterfactual scenario is not to outline a realistic path, but to isolate the greenhouse gasses associated with the growth needed to end poverty. If we were to use more realistic growth rates in the counterfactual scenario, some countries, such as India, would reach the level of growth needed to end poverty in the counterfactual scenario, and the growth needed to end poverty would be zero. This would be relevant if we wanted to capture the emissions associated with the growth needed to end poverty in countries that aren’t on track to end it by 2050. Yet we are interested in all growth needed to end poverty. For that, a counterfactual scenario with no growth is necessary. We have tried to change the language to make the confusion less likely to arise.

- Last paragraph p.7, regarding my earlier comment, why just poor countries??

We hope the earlier answer helps explain why we just considered poor countries here. We would be happy to clarify further as needed.

- First paragraph p.8, would say “is unlikely” instead of “cannot be expected “

Good point, now corrected

- Final paragraph, p.8: again when saying “and the need to eradicate extreme poverty cannot be used as a justification for reducing the world’s climate ambitions” would specify ‘and vice versa’ too

We have now edited the sentences to reflect this point.

- Final sentence, “Our analysis faces important limitations.” This is an extremely abrupt and strange concluding sentence. I agree there are important limitations, a number of which I touched up on in my general comments. You need to elaborate substantially here

Certainly, we have now added several paragraphs to discuss the limitations in greater detail.

Reviewer Reports on the First Revision:

Referees' comments:

Referee #1 (Remarks to the Author):

The revision of this manuscript helped address some of my concerns and the resulting manuscript is more convincing. In the end the growth needed to get to where this paper is focused are not all that high. Sub-Saharan Africa would need to grow a lot but would still be a long ways behind the rest of the world. Other parts of the developing world (in particular South Asia) would have to increase their GDP per capita by something 20 or 30 percent. These relatively modest increases in GDP per capita overall would lead to relatively modest decreases in population and relative modest increases in emissions. At the end of the day I believe that conclusion.

Referee #2 (Remarks to the Author):

The authors have made substantial efforts to address my comments and the analysis in the paper is stronger as a result.

I am satisfied with how they have addressed the comments on country-specific linear time-trends (in the consumption-GDP regression) and confidence intervals for the main results. The added discussion on the limitations of the basic methodology and the distribution neutrality assumption is also a welcome addition.

However, I continue to be concerned about the framing of the overall result of the paper. While I do appreciate the framing in the concluding discussion (i.e. on lines 260-262: "Since bringing poverty to 0% is disproportionately more growth intensive, our results diverge from prior studies as we move closer to 0%."), the initial sentences in both the abstract and summary paragraph continue to give a misleading impression about the similarity between this study's results and those of previous studies.

In particular, I find the use of the word "eradicate" to be inappropriate, since this study and previous ones mean two different things when using this term in relation to poverty. The word "alleviate" seems more appropriate to me for this study.

The authors argue that the 0% target poverty rate is inappropriate for the two distinct reasons of transitory poverty and measurement error. While these reasons are not without merit, I would prefer to see the 0% target poverty rate retained in the Extended Data Figure 12, because 1) it is the benchmark in the other studies, and 2) there appear to exist countries that today have 0% of the population under the \$2.15 threshold, and even more countries in the 0%-1% range (<https://data.worldbank.org/indicator/SI.POV.DDAY>).

Referee #3 (Remarks to the Author):

I've reviewed the revised manuscript and the authors' responses to my comments. Given the expanded discussion of the limitations of the analysis, I'm satisfied with these revisions and happy to support publication.

Author Rebuttals to First Revision:

Point-by-point response to referee comments

Referee #1 (Remarks to the Author):

The revision of this manuscript helped address some of my concerns and the resulting manuscript is more convincing. In the end the growth needed to get to where this paper is focused are not all that high. Sub-Saharan Africa would need to grow a lot but would still be a long ways behind the rest of the world. Other parts of the developing world (in particular South Asia) would have to increase their GDP per capita by something 20 or 30 percent. These relatively modest increases in GDP per capita overall would lead to relatively modest decreases in population and relative modest increases in emissions. At the end of the day I believe that conclusion.

Thank you for your assessment and support.

Referee #2 (Remarks to the Author):

The authors have made substantial efforts to address my comments and the analysis in the paper is stronger as a result.

I am satisfied with how they have addressed the comments on country-specific linear time-trends (in the consumption-GDP regression) and confidence intervals for the main results. The added discussion on the limitations of the basic methodology and the distribution neutrality assumption is also a welcome addition.

However, I continue to be concerned about the framing of the overall result of the paper. While I do appreciate the framing in the concluding discussion (i.e. on lines 260-262: "Since bringing poverty to 0% is disproportionately more growth intensive, our results diverge from prior studies as we move closer to 0%."), the initial sentences in both the abstract and summary paragraph continue to give a misleading impression about the similarity between this study's results and those of previous studies.

- Thank you for raising this. In the process of cutting the abstract and summary paragraph into maximum 200 words, we have removed the sentence that gives the misleading comparison to previous studies. In its place, we have expanded the discussion comparing our results to prior studies, which now reads: "More subtle choices, such as the target year for ending poverty (2050) and the target poverty rate (3%) also matter for the results (Extended Data Figure 9a-b). Prior studies, which focused on reducing poverty to 0% without impacting consumption of non-poor, found this to increase global emissions by 2.8%, 1.6-2.1%, and 1.9%. Using a 0% target is not meaningful in our framework because it makes the growth needed to end poverty dependent on the poorest households in the country. Because of transitory poverty (e.g., due to health shocks) and because of challenges in measuring consumption of very poor people, using a 0% target makes our results less reliable and unstable and they diverge from prior studies as we move closer to 0%. Our framing is also less relevant for very low poverty rates: to eradicate the last pockets of extreme poverty in a country, social protection schemes and redistribution plays a stronger role than economy-wide economic growth. In our framework, such transfers are represented as a reduction in inequality, whose effects on emissions have been explored earlier."

In particular, I find the use of the word "eradicate" to be inappropriate, since this study and previous ones mean two different things when using this term in relation to poverty. The word "alleviate" seems more appropriate to me for this study.

- This is a fair point. We have now removed all references to "eradicate" poverty and have replaced most of them with "alleviate." This includes in a subtitle which now reads, "The emissions implications of poverty alleviation" and our main scenario, which now is called the 'poverty-alleviation scenario' rather than the 'poverty-eradication scenario.'" Note that, in a certain cases, based on what made most substantive and semantical sense, instead of using 'alleviation', we refer to the 3% target or have used the word "reducing" or "ending."
- Note we have also removed references to "eliminating poverty."

The authors argue that the 0% target poverty rate is inappropriate for the two distinct reasons of transitory poverty and measurement error. While these reasons are not without merit, I would prefer to see the 0% target poverty rate retained in the Extended Data Figure 12, because 1) it is the benchmark in the other studies, and 2) there appear to exist countries that today have 0% of the population under the \$2.15 threshold, and even more countries in the 0%-1% range (<https://data.worldbank.org/indicator/SI.POV.DDAY>).

- All good points, we have added the 0% target back to Extended Figure 12.

Referee #3 (Remarks to the Author):

I've reviewed the revised manuscript and the authors' responses to my comments. Given the expanded discussion of the limitations of the analysis, I'm satisfied with these revisions and happy to support publication.

Thank you for your support and comments.